# A slit-diaphragm-associated protein network for dynamic control of renal filtration

Maciej K. Kocylowski [1,8], Hande Aypek[2,8], Wolfgang Bildl[1], Martin Helmstädter[3], Philipp Trachte[3], Bernhard Dumoulin[2], Sina Wittösch[2], Lukas Kühne [3], Ute Aukschun[3], Carolin Teetzen[3], Oliver Kretz[2], Botond Gaal [1,4], Akos Kulik[1], Corinne Antignac [5], Geraldine Mollet [5], Anna Köttgen [6], Burulca Göcmen[6], Jochen Schwenk [1], Uwe Schulte [1], Tobias B. Huber [2,9] ✉, Bernd Fakler [1,7,9] ✉ & Florian Grahammer[2,9] ✉

The filtration of blood in the kidney which is crucial for mammalian life is determined by the slit-diaphragm, a cell-cell junction between the foot processes of renal podocytes. The slit-diaphragm is thought to operate as final barrier or as molecular sensor of renal filtration. Using high-resolution proteomic analysis of slit-diaphragms affinity-isolated from rodent kidney, we show that the native slit-diaphragm is built from the junction-forming components Nephrin, Neph1 and Podocin and a co-assembled high-molecular weight network of proteins. The network constituents cover distinct classes of proteins including signaling-receptors, kinases/phosphatases, transporters and scaffolds. Knockout or knock-down of either the core components or the selected network constituents tyrosine kinase MER (MERTK), atrial natriuretic peptide-receptor C (ANPRC), integral membrane protein 2B (ITM2B), membrane-associated guanylate-kinase, WW and PDZ-domain-containing protein1 (MAGI1) and amyloid protein A4 resulted in target-specific impairment or disruption of the filtration process. Our results identify the slit-diaphragm as a multi-component system that is endowed with context-dependent dynamics via a co-assembled protein network.

Renal filtration is fundamental for mammalian life as it effectively removes metabolic waste products and toxins from the blood and rapidly clears any (extra)-loads of water and electrolytes crucial for homeostasis. Such effective excretion is only enabled by an elaborate filtration unit, the glomerulus, which allows for the passage of large amounts of blood required for effective filtration of small solutes and which, at the same time, largely retains macromolecules such as albumin and prevents clogging over long periods of life time[1,2].

The centerpiece of the glomerular unit is the three-layered filter that is made up of the fenestrated capillary endothelium, the glomerular basement membrane (GBM), and the slit-diaphragm (SD), a proteinaceous junction between the foot processes (FP) of neighboring podocytes[3,4]. The latter are glomerulus-specific cells that entirely engulf the capillaries with their FPs and are attached to the GBM by adhesion complexes[4]. In line with this serial arrangement, GBM and the SD have been assumed to operate as the decisive barrier of the filtration process[5,6]

While the architecture and physiology of glomerular filtration have been extensively studied in the past, molecular understanding of the filtration process is still limited, as is the current knowledge on the SD and its role therein[4,7]. Studies of rare genetic diseases have identified a number of gene products that are important for glomerular physiology[8]. Two of these genes, *NPHS1* and *NPHS2*, encoding proteins Nephrin and Podocin, are abundantly expressed in podocytes and were shown to cause massive proteinuria when mutated[9–11].

Subsequent analysis unveiled Nephrin and Podocin as core components of the SD, and also identified Neph1, a homolog of Nephrin, as another SD component[12–14]. Improved microscopy techniques, in particular cryoelectron-tomography, showed that Nephrin and Neph1 span the slit between neighboring FPs and implied that this slit-covering layer does most likely not operate as a filtration barrier[7,15]. This leaves the question about the primary function of the SD that covers about 1/6 of the glomerular capillary area open.

Key in the pursuit of this question is a rigorous analysis of the building blocks of the SD from native podocytes as it is enabled by quantitative high-resolution proteomics established for studying membrane proteins and protein complexes[16,17]. This approach that combines affinity-purification of target proteins with nano-flow tandem mass spectrometry (nano-LC-MS/MS) was successfully used to unravel the subunit composition and complexity of a number of ion channels, transporters, and receptors[18–25].

Here, we combined our proteomic workflow with functional investigations and gene knockout and knockdown to investigate the molecular appearance and protein composition of the SD in rodent podocytes and to assess the functional significance of the building blocks identified by the proteomic approach.

## Results

### Interactome of the SD core constituents Nephrin and Neph1

For comprehensive and quantitative insight into the molecular composition of the rodent SD, we set out to perform multi-epitope affinity purifications (meAPs) with antibodies (ABs) targeting the known core components Nephrin, Neph1, and Podocin. As source material, we used membrane fractions prepared from glomeruli of both rat and mouse kidney, where all core constituents were highly enriched (Supplementary Fig. 1, Methods).

We first tested a series of commercial and custom-made ABs with distinct epitopes on the target proteins (Supplementary Fig. 2) for specificity and efficiency of purification and probed various detergent buffers for their solubilization characteristics. For Nephrin and Neph1 these efforts identified three different ABs that were suitable for APs (Fig. 1A, Supplementary Fig. 2), and one of them directed against the extracellular domain of Nephrin enabled immuno-staining of SDS-digested freeze-fracture replicas prepared from isolated glomeruli. Figure 1B illustrates a representative EM-pictogram of podocyte pedicles showing the exclusive distribution of the Nephrin-labeling immuno-gold particles to the extracellular surface (E-face) of the slit between interdigitating FPs similar to results from STED-microscopy[26].

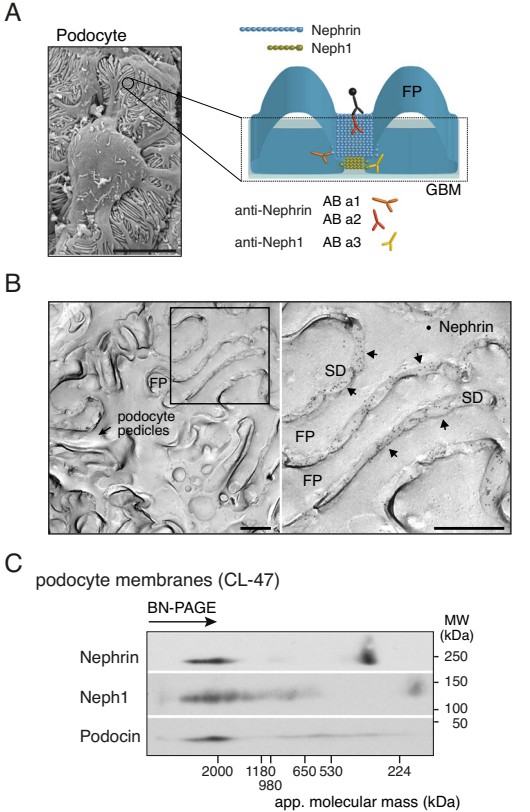

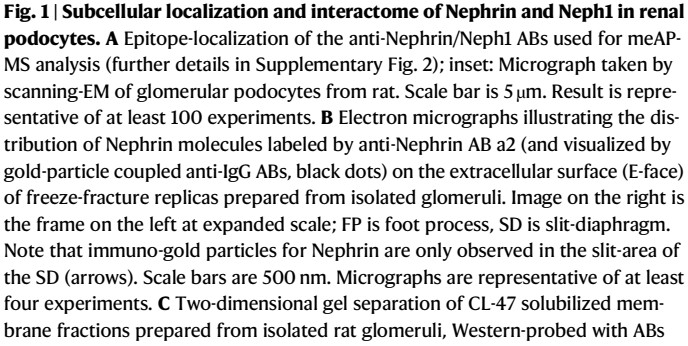

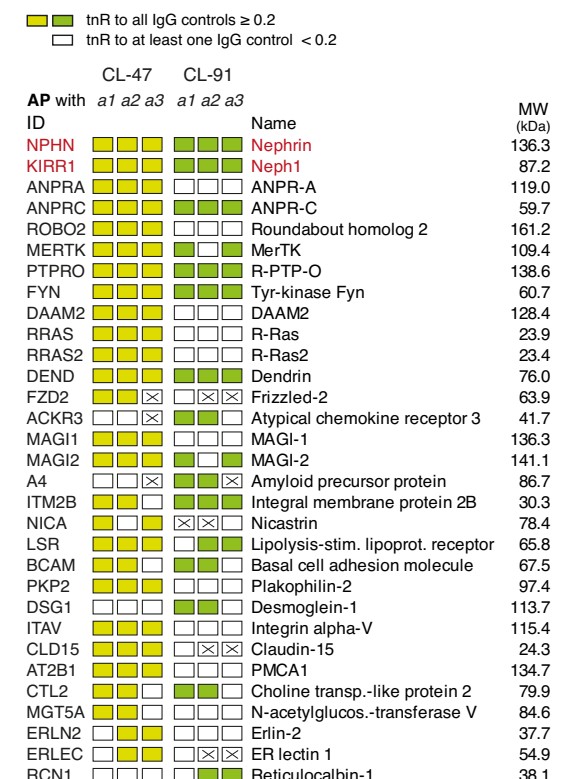

**Fig. 1 | Subcellular localization and interactome of Nephrin and Neph1 in renal podocytes. A** Epitope-localization of the anti-Nephrin/Neph1 ABs used for meAP-MS analysis (further details in Supplementary Fig. 2); inset: Micrograph taken by scanning-EM of glomerular podocytes from rat. Scale bar is 5 μm. Result is representative of at least 100 experiments. **B** Electron micrographs illustrating the distribution of Nephrin molecules labeled by anti-Nephrin AB a2 (and visualized by gold-particle coupled anti-IgG ABs, black dots) on the extracellular surface (E-face) of freeze-fracture replicas prepared from isolated glomeruli. Image on the right is the frame on the left at expanded scale; FP is foot process, SD is slit-diaphragm. Note that immuno-gold particles for Nephrin are only observed in the slit-area of the SD (arrows). Scale bars are 500 nm. Micrographs are representative of at least four experiments. **C** Two-dimensional gel separation of CL-47 solubilized membrane fractions prepared from isolated rat glomeruli, Western-probed with ABs

against the indicated proteins. Apparent molecular mass (native PAGE, 1st dimension) and molecular weight (denaturing SDS-PAGE, 2nd dimension) are indicated; gel separations were repeated twice with similar results. Note incorporation of all three SD core constituents into high-molecular-mass complexes of ~2 MDa. **D** Table summarizing the results of the finally annotated constituents of the Nephrin/Neph1 interactome in all six APs with the indicated ABs (Supplementary Fig. 2; AB a3 refers to anti-Neph1 a1). Each column illustrates the results of the indicated target AP controlled by three target-unrelated APs. Color-coding symbolizing evaluation by target-normalized ratios (tnR) as follows: yellow/green: tnR against all thee controls >0.2, white: at least one tnR below the threshold of 0.2, crossed: protein was not identified in the respective target AP. ID refers to the Swiss-Prot entry of the respective protein. MW of all proteins is indicated on the right.

Table (Fig. 1D). tnR to all IgG controls ≥ 0.2 (yellow/green); tnR to at least one IgG control < 0.2 (white). AP with CL-47 (a1 a2 a3) and CL-91 (a1 a2 a3):

| ID | Name | MW (kDa) |
|---|---|---|
| NPHN | Nephrin | 136.3 |
| KIRR1 | Neph1 | 87.2 |
| ANPRA | ANPR-A | 119.0 |
| ANPRC | ANPR-C | 59.7 |
| ROBO2 | Roundabout homolog 2 | 161.2 |
| MERTK | MerTK | 109.4 |
| PTPRO | R-PTP-O | 138.6 |
| FYN | Tyr-kinase Fyn | 60.7 |
| DAAM2 | DAAM2 | 128.4 |
| RRAS | R-Ras | 23.9 |
| RRAS2 | R-Ras2 | 23.4 |
| DEND | Dendrin | 76.0 |
| FZD2 | Frizzled-2 | 63.9 |
| ACKR3 | Atypical chemokine receptor 3 | 41.7 |
| MAGI1 | MAGI-1 | 136.3 |
| MAGI2 | MAGI-2 | 141.1 |
| A4 | Amyloid precursor protein | 86.7 |
| ITM2B | Integral membrane protein 2B | 30.3 |
| NICA | Nicastrin | 78.4 |
| LSR | Lipolysis-stim. lipoprot. receptor | 65.8 |
| BCAM | Basal cell adhesion molecule | 67.5 |
| PKP2 | Plakophilin-2 | 97.4 |
| DSG1 | Desmoglein-1 | 113.7 |
| ITAV | Integrin alpha-V | 115.4 |
| CLD15 | Claudin-15 | 24.3 |
| AT2B1 | PMCA1 | 134.7 |
| CTL2 | Choline transp.-like protein 2 | 79.9 |
| MGT5A | N-acetylglucos.-transferase V | 84.6 |
| ERLN2 | Erlin-2 | 37.7 |
| ERLEC | ER lectin 1 | 54.9 |
| RCN1 | Reticulocalbin-1 | 38.1 |

In fact, the surface view provided by the freeze-fractures directly demonstrated the layer-like arrangement of Nephrin molecules covering the entire area of the slit (Fig. 1B[7]). Probing of detergent buffers with distinct stringencies showed that the previously established buffer systems CL-47 (mild stringency) and CL-91 (intermediate stringency) effectively solubilized the three core proteins, albeit at distinct efficiency (Supplementary Fig. 3A). Interestingly, separation by native gel electrophoresis (BN-PAGE) demonstrated that all three SD core constituents Nephrin, Neph1, and Podocin are integrated into high-molecular-weight assemblies of ~1.5–3 MDa in mass, and that these macro-molecular assemblies are well preserved by both of the aforementioned buffer systems (Fig. 1C, Supplementary Fig. 3B). Importantly, these macro-molecular assemblies do not focus on one sharp population suggesting that SD core constituents might be involved in the formation of multiple distinct sub-complexes. Moreover, Podocin seemed to participate in additional assemblies of somewhat smaller molecular mass (~250–650 kDa; Fig. 1C, lower panel).

To resolve the subunit composition of the high-molecular-mass assemblies, we first focused on the slit-spanning Nephrin and Neph1 and, thus, performed APs with the identified anti-Nephrin/Neph1 ABs and three target-unrelated ABs (or IgGs as negative controls) on glomerular membrane fractions solubilized with either CL-buffer. The resulting elutions were analyzed by high-resolution nano-LC-MS/MS and the respective MS data were evaluated by our label-free quantification procedure that is based on peak-volume (PV) determination and offers a linear dynamic range of more than four orders of magnitude[16,24,25,27].

The results of these analyses showed that both Nephrin and Neph1 were effectively purified as indicated by large PV values and extensive relative coverage of the primary sequence(s) by MS/MS-identified peptides (64% and 69%, respectively; Fig. 1D, Supplementary Fig. 4; details on all aspects of MS-analyses are given in Supplemental Material, Supplementary Data 1). Noteworthy, both proteins mutually co-purified each other, while Podocin was not specifically retained in either of the three anti-Nephrin/Neph1 APs (Fig. 1C, Supplementary Data 1). Moreover, the effective isolation enabled MS-investigation of protein phosphorylation (without further enrichment) identifying several modified residues in the cytoplasmic C-termini of both Nephrin and Neph1 (Supplementary Fig. 4). Strikingly, in Nephrin phosphorylation appeared clustered in a stretch of three serine/threonine residues within an acidic (glutamine-rich) domain[28] adjacent to the single transmembrane segment. Identification of multiple double and triple phosphorylated peptides from that region (Supplementary Fig. 4, Supplementary Table 1) suggested a high degree of phosphorylation at steady state.

All other proteins retrieved in the anti-Nephrin/Neph1 APs and identified in our MS analyses (exceeding the detection threshold, Methods, Supplementary Fig. 5) were evaluated for specificity of their co-purification with the respective target, and subsequently for consistency of their specific retrieval. Specificity of co-purification (of any protein) was determined via relative abundance ratios using the target-unrelated ABs/IgGs as negative controls (or background) and the abundance ratios determined for Nephrin or Neph1 as respective references (target-normalized ratio (tnR) for direct comparability, Methods, Supplementary Fig. 5). Consistency was defined as the number of specific co-purifications (of any protein) across the anti-Nephrin/Neph1 APs performed with the CL-47 or CL-91 solubilized membrane fractions and served to eliminate potential errors (false positives) resulting from peculiarities of individual ABs and/or APs[16,20,22,23]. A tnR threshold value of ≥0.2 and the request for specific co-purification in at least two anti-Nephrin/Neph1 APs finally identified an ensemble of 29 proteins (out of 1750 proteins detected) that robustly co-purified with Nephrin and Neph1 and thus defined the bona fide interactome (or proteome) of these slit-spanning SD core constituents (Fig. 1D, Table 1). With exception of tyrosine kinase Fyn

and proteins Dendrin, ROBO2, and MAGI1,2[29–33], none of the identified interactors have yet been associated with Nephrin or Neph1 in public databases or literature. Eight interactors have previously been ascribed to podocyte function/biology (mostly through their gene knockout, underlined in Table 1).

Detailed inspection of the interactome showed a number of intriguing features (Table 1): First, most of the interactome constituents are membrane proteins of different classes (majority with a single membrane-spanning domain); the others are soluble proteins with documented membrane-attachment(s) via lipid anchors or known binding to membrane proteins (through WW or PDZ-domain(s)). Second, the molecular mass of the individual interactome constituents ranged from ~25–160 kDa thus indicating that polyvalent interactions are required to finally generate the high-molecular-mass assemblies observed in native gels (Figs. 1C, D, Supplementary Fig. 3B). Third, similar to Nephrin and Neph1, several interactors exhibit extended extracellular domains comprising Ig- or Ig-like domains implicated in transcellular protein-protein interaction(s). Fourth, according to currently annotated (or established) primary function(s) the interactome constituents cover several aspects of cell biology, most prominently signaling at and across the plasma membrane (via receptors, kinases/phosphatases, GTPases) and cellular junction(s) (cell adhesion, scaffolding). Fifth, for several of the Nephrin/Neph1-interactors primary function(s) remain as yet unknown or only partially resolved. Sixth, and noteworthy, parts of the interactome including proteins PTPRO[34,35], MAGI1/2[36–38], ROBO2[39,40], Dendrin, as well as the amyloid-beta precursor protein A4 and its interactor ITM2B[23] share expression in the brain where they are predominantly localized to the synaptic compartment.

Together, these results indicated that the SD core components Nephrin and Neph1 are associated with a variety of membrane and membrane-associated proteins that likely combine to generate high-molecular-weight networks associated with the SD and that are mostly known for signaling processes and formation of a cellular junction(s).

## Interactome of the podocyte-specific SD component Podocin

The SD-associated networks were further pursued by next applying the meAP approach to the cytoplasmic SD anchor-protein Podocin, a hairpin-like protein firmly attached to the cytoplasmic leaflet of the membrane[41,42], as a primary target.

Similar to Nephrin/Neph1, AB testing led to the identification of two AP-suited ABs with distinct epitopes located close to the N- and C-termini of the protein (Fig. 2A, Supplementary Fig. 2). Moreover, one of these ABs could be used to determine the subcellular localization of Podocin in freeze-fracture replicas (Fig. 2B). In contrast to Nephrin, Podocin-staining displayed a broad distribution over the cytoplasmic membrane face (P-face) of the podocyte FPs. Thus, ~40% of the protein were located close to the plasma membrane right underneath the slit (distance to the membrane ≤60 nm) in line with previous observations[13], while the remainder of Podocin appeared distributed over the entire area of the sole of the FPs (Fig. 2B).

For meAPs, the two identified anti-Podocin ABs (Supplementary Fig. 2) were used on CL47- and CL91-solubilized glomerular membrane fractions obtained either from wildtype mice, or from mice with an induced knockout of Podocin (Methods). The latter served as a stringent negative control for determining the specificity of co-purification which can be visualized by a t-SNE (t-distributed stochastic neighbor embedding) plot of the tnR-values determined for all proteins in APs from wildtype and target knockout membranes (Fig. 2C, CL-47). Accordingly, the abundantly retrieved primary target Podocin (large PV values, relative sequence coverage of 74%; MS-details Supplemental Material, Supplementary Data 2 and Supplementary Fig. 4 together with the conserved C-terminal phosphorylation site S362[28]) was closely neighbored by a number of proteins in the t-SNE plot (Fig. 2C, highlighted by colored squares) which shared a tnR-value for specific

**Table 1 | Constituents of the Nephrin/Neph1 interactome determined by meAP-MS, together with annotation of subcellular localization and primary function(s) indicated in literature and public databases**

| ID | Acc. no. | Name | Topology/localization | Function |
|---|---|---|---|---|
| NPHN | Q9R044 | Nephrin | 1 TM | Cell adhesion, scaffold |
| KIRR1 | Q6X936 | Neph1 | 1 TM | Cell adhesion, scaffold |
| ANPRA* | P18910 | ANPRA | 1 TM | Receptor-mediated signaling, guanylyl cyclase |
| ANPRC* | P41740 | ANPRC | 1 TM | Receptor-mediated signaling |
| ROBO2 | Q7TPD3 | Roundabout homolog 2 | 1 TM | Receptor-mediated signaling |
| MERTK* | P57097 | MerTK | 1 TM | Receptor kinase |
| PTPRO* | E9Q612 | R-PTP-O | 1 TM | Receptor phosphatase, cell adhesion |
| FYN | Q62844 | Tyr-kinase Fyn | Membrane anchored (lipidation) | Kinase |
| DAAM2* | Q80U19 | DAAM2 | Cytoplasmic/membrane associated | Signaling |
| RRAS* | D3Z8L7 | R-Ras | Membrane anchored (lipidation) | Signaling, small GTPase |
| RRAS2* | P62071 | R-Ras2 | Membrane anchored (lipidation) | Signaling, small GTPase |
| DEND | P50617 | Dendrin | Membrane associated/nuclear | Signaling, apoptosis |
| FZD2* | Q08464 | Frizzled-2 | 7 TMs | GPCR |
| ACKR3* | O89039 | Atypical chemokine receptor 3 | 7 TMs | GPCR |
| MAGI1 | Q6RHR9 | MAGI-1 | membrane associated | Scaffold |
| MAGI2 | O88382 | MAGI-2 | membrane associated | Scaffold |
| A4* | P08592 | Amyloid precursor protein | 1 TM | APP, cell adhesion |
| ITM2B* | Q5XIE8 | Integral membrane protein 2B | 1 TM | APP regulation |
| NICA* | Q8CGU6 | Nicastrin | 1 TM | APP regulation |
| LSR* | Q9WU74 | Lipolysis-stimulated lipoprotein receptor | 1 TM | Cell adhesion |
| BCAM* | Q9ESS6 | Basal cell adhesion molecule | 1 TM | Cell adhesion |
| PKP2* | Q99959 | Plakophilin-2 | Membrane associated | Cell adhesion |
| DSG1* | Q02413 | Desmoglein-1 | 1 TM | Cell adhesion |
| ITAV* | P43406 | Integrin alpha-V | 1 TM | Cell adhesion |
| CLD15* | D3ZQJ0 | Claudin-15 | 4 TMs | Cell adhesion |
| AT2B1* | P11505 | PMCA1 | 10 TMs | Transporter |
| CTL2* | B4F795 | Choline transporter-like protein 2 | 10 TMs | Transporter |
| MGT5A* | Q08834 | N-acetyl-glucosaminyl-transferase V | 1 TM | N-acetyl-glucosaminyltransferase |
| ERLN2* | B5DEH2 | Erlin-2 | ER, 1 TM | ERAD |
| ERLEC* | Q96DZ1 | ER lectin 1 | ER lumen | ERAD |
| RCN1* | Q05186 | Reticulocalbin-1 | ER lumen | Calcium binding |

Accession codes refer to the UniProtKB/Swiss-Prot database; underlining refers to previous report(s) on podocyte-related function(s), asterisks denote proteins that have not been reported to interact with Nephrin or Neph1 before.

co-purification of >0.25. This threshold value together with the consistency request of specific co-purification with both anti-Podocin ABs in either of two solubilization conditions finally defined the interactome of Podocin (Fig. 2D, Table 2).

Structurally, this interactome implies robust association of Podocin with the two slit-spanning SD components Nephrin and Neph1, in line with previous work and directly visualized by the close proximity in the t-SNE plot (Fig. 2C). In addition, several other constituents of the Neprin/Neph1 interactome were identified as specific partners of Podocin including tyrosine kinase Mer (MERTK), the single-span membrane proteins MAGI2, PTPRO, and ANPRA[43], as well as the transporter CTL2 (Fig. 1D, Table 1). Beyond these SD components, however, the Podocin interactome contained a number of additional proteins (Fig. 2D, Table 2), in line with the distinct subcellular localization of Podocin compared to the exclusive localization of Nephrin/Neph1 to the slit between podocyte FPs (Fig. 2B). Interestingly, the majority of these Podocin-specific interactors are either members of different (sub)-families of solute-carrier (SLC)-type transporters[44] or other transporting systems (ABC-A9, MFSD1) and scaffolding proteins (tetraspanin CD81). The individual transporters differ in their substrate(s) ranging from organic anions/cations to amino acids or vitamins and, as yet, none of them has been related to Podocin in public databases and literature.

Together, these interactome data strongly suggest that Podocin serves as a more universal cytoplasmic anchor targeting not only the SD and its associated protein network, but also integral membrane proteins with diverse functions.

### Significance of the core constituents for renal filtration

Given the distinct composition of their interactomes, we next went to investigate the properties and significance of the three SD core components in the context of the rodent kidney with particular emphasis on their stability and time course of protein turnover, as well as their impact on filtration and podocyte morphology.

For this purpose, we generated mice where either of the three genes *Nphs1* (Nephrin), *Nphs2* (Podocin), and *Neph1* (Neph1) could be specifically knocked-out in podocytes upon doxycycline administration (Methods) and analyzed them via MS-based protein biochemistry, conventional and fluorescence-based microscopy, as well as time-dependent measurements of albumin content in the urine (Fig. 3, Supplementary Figs. 6–8). Generation of these inducible and podocyte-specific knockout animals ($Nphs1^{\Delta iPod}$, $Nphs2^{\Delta iPod}$, $Neph1^{\Delta iPod}$, Methods) was necessary since constitutive *Nphs1*, *Nphs2*, and *Neph1* knockout mice all die perinatally thus preventing in-depth studies of dynamics[11,12,45]. Important to note, all inducible models used the same promotor for gene deletion and were done in

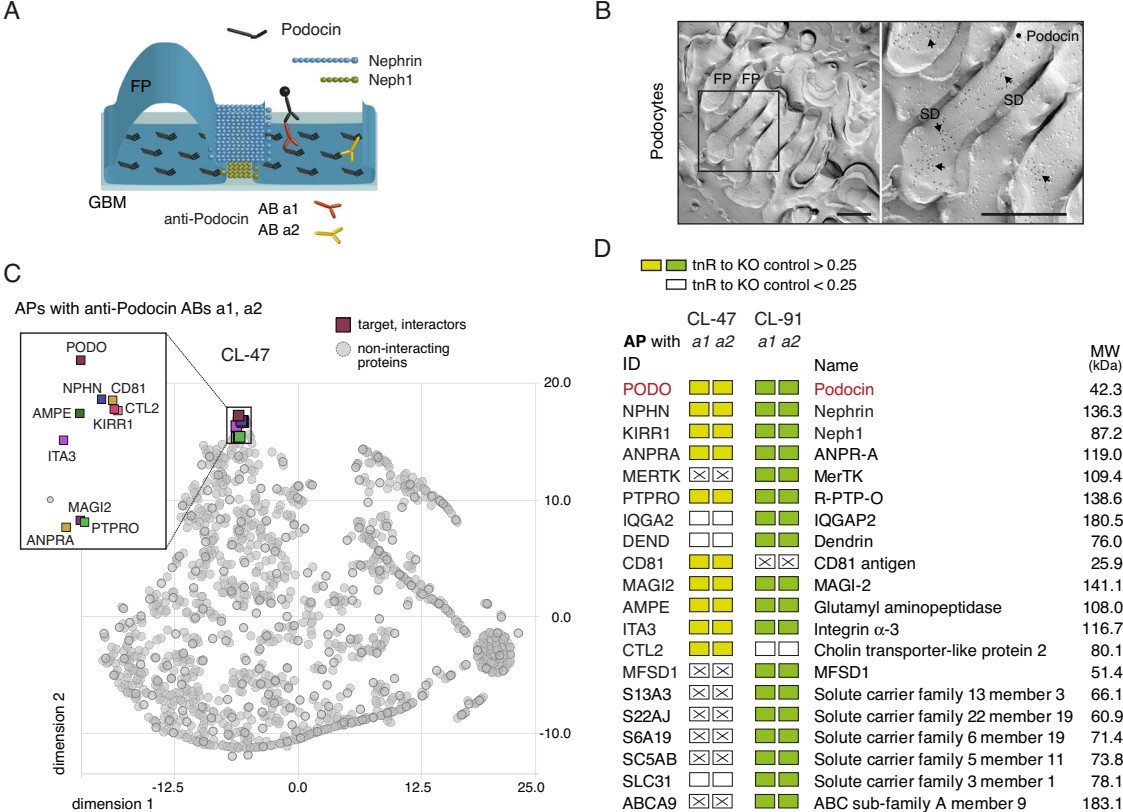

**Fig. 2 | Subcellular distribution and interactome of Podocin in renal podocytes.**
**A** Epitope-localization of the anti-Podocin ABs used for meAP-MS analysis (further details in Supplementary Fig. 2). **B** Electron micrographs showing the distribution of Podocin proteins labeled by anti-Podocin AB a1 (and visualized by gold-particle coupled anti-IgG ABs, black dots) on the cytoplasmic side (P-face) of the plasma membrane in freeze-fracture replicas prepared from isolated glomeruli. Image on the right is the frame on the left at an expanded scale. Note that immuno-gold particles for Podocin (arrows) are observed close to the slit-facing membrane, as well as on the sole of the podocyte foot processes (FPs). Scale bars are 500 nm. Micrographs are representative of at least four experiments. **C** t-SNE plot of tnR-values determined for all proteins identified in anti-Podocin APs from CL-47

solubilized membrane fractions of isolated mouse glomeruli (using KO source materials as negative controls). Podocin and all proteins clustered by t-SNE analysis close to the target are indicated by colored squares and shown at an extended scale. **D** Table summarizing the results of the finally annotated constituents of the Podocin interactome in all APs with the indicated ABs (Supplementary Fig. 2). Each column illustrates the results of the indicated anti-Podocin AP controlled by the target KO. Color-coding symbolizing evaluation by target-normalized ratios (tnR, AP versus control) as follows: yellow/green: tnR against KO control >0.25, white: tnR below the threshold of 0.25, crossed: protein was not identified in the respective target AP. MW of all proteins is indicated on the right. ID refers to the Swiss-Prot entry of the respective protein.

the identical genetic background strain (C57Bl6/NCrl) in the same animal facility.

Analysis of mRNA abundance one week after initiating doxycycline-induction confirmed the decrease in target-gene expression (to $21 \pm 4\%$ of control (mean $\pm$ SEM) for *Nphs2* ($n = 6$), $12 \pm 3\%$ ($n = 6$) *Nphs1* and $28 \pm 13\%$ ($n = 5$) for *Neph1*, respectively) which remained stable (or decreased further) thereafter as indicated by the 3-week post induction results (Fig. 3A–C, left panel). For Podocin, the decreased mRNA levels were closely followed by a robust and time-dependent decrease in protein levels which was visualized by immuno-staining of glomerular podocytes one and three weeks after induction (Fig. 3D), and which was confirmed by MS-based protein quantification in total membrane fractions (-95% reduction compared to controls three weeks post-induction, Supplementary Fig. 7). A decrease in protein amount was also observed in MS analyses of both Nephrin and Neph1 knockouts, although the time course appeared markedly delayed and efficiency reached values of ~45% three and four weeks after doxycycline-induction, respectively (Supplementary Fig. 7).

As a consequence of the knockout-induced reduction in protein expression, the renal filtration process assessed by determination of the albumin/creatinine ratio in the urine was altered in a target-specific manner (Fig. 3E). Thus, *Nphs2^ΔiPod^* animals started to display albumi-nuria (albumin/creatinine ratio >2) by day 14, which rapidly increased

to massive nephrotic range values over the following days (albumin/creatinine ratios >200 at day 21, Fig. 3C, right panel). A similarly early onset of albuminuria was observed with *Neph1^ΔiPod^* mice, while in *Nphs1^ΔiPod^* animals the albumin/creatinine ratios of >2 were not observed before day 68 after induction (Fig. 3C, right panels, Fig. 3E). Nephrotic range values were determined at days 28 and 124 for Neph1 and Nephrin knockout, respectively.

Prompted by these marked differences in the onset of albumi-nuria, we extended the protein quantification in total membrane fractions prepared from KO animals to the additionally identified constituents of the Nephrin/Neph1 and Podocin interactomes (Figs. 1D, 2D, Supplementary Data 1, 2). The respective data uncovered several interesting findings related to the alterations in SD/protein stability upon knockout of the SD core components (Supplementary Fig. 7). First, in Podocin and Neph1 KO mice, the amounts of Dendrin, MAGI2, and FYN were markedly reduced (mirroring the targets) at early time points (2 and 4 weeks post-induction), while in Nephrin KOs their amounts decreased only 12 weeks after KO induction (Supplementary Fig. 7A–C). Second, Dendrin, MAGI2 and PTPRO, and, to a lesser extent, MERTK, ANPRA, and FYN displayed direct dependence on the expression of the SD core constituents, while the other interactors appeared largely independent (Supplementary Fig. 7D). Third, severe albuminuria in all three cases appeared to be correlated with the decrease in Neph1. Finally, in Nephrin mice, induction of the KO

**Table 2 | Constituents of the Podocin interactome determined by meAP-MS, together with annotation of subcellular localization and primary function(s) indicated in literature and public databases**

| ID | Acc. No. | Name | Topology/localization | Function |
|---|---|---|---|---|
| PODO | Q91X05 | Podocin | 1 IM | Scaffold |
| NPHN | Q9QZS7 | Nephrin | 1 TM | Cell adhesion, signaling |
| KIRR1 | Q80W68 | Neph1 | 1 TM | Cell adhesion, signaling |
| ANPRA* | P18293 | ANPRA | 1 TM | Receptor mediated signaling, guanylyl cyclase |
| MERTK* | Q60805 | MerTK | 1 TM | Receptor kinase |
| PTPRO* | E9Q612 | R-PTP-O | 1 TM | Receptor phosphatase, cell adhesion |
| IQGA2* | Q3UQ44 | IQGAP2 | cytoplasmic/membrane associated | Signaling |
| DEND | Q80TS7 | Dendrin | membrane associated/nuclear | Signaling, apoptosis |
| CD81* | P35762 | CD81 antigen | 4 TMs | Scaffold/signaling |
| MAGI2 | Q9WVQ1 | MAGI-2 | membrane associated | Scaffold |
| AMPE* | P16406 | Glutamyl aminopeptidase | 1 TM | Glutamyl aminopeptidase |
| ITA3* | Q62470 | Integrin alpha-3 | 1 TM | Cell adhesion |
| CTL2* | Q8BY89 | Choline transporter-like protein 2 | 10 TMs | Transporter |
| MFSD1* | Q9DC37 | MFSD1 | 11 or 12 TMs | Transporter |
| S13A3* | Q91Y63 | Solute carrier family 13 member 3 | 11 TMs/1 IM | Transporter |
| S22AJ* | Q8VCA0 | Solute carrier family 22 member 19 | 11 TMs | Transporter |
| S6A19* | Q9D687 | Solute carrier family 6 member 19 | 12 TMs | Transporter |
| SC5AB* | Q8K0E3 | Solute carrier family 5 member 11 | 14 TMs | Transporter |
| SLC31* | Q91WV7 | Solute carrier family 3 member 1 | 1 TM | Transporter |
| ABC-A9* | Q8K449 | ABC sub-family A member 9 | 14 TMs | Transporter |

Accession codes refer to the UniProtKB/Swiss-Prot database; underlining refers to previous report(s) on podocyte-related function(s), asterisks denote proteins that have not been reported to interact with Podocin before.

predominantly reduced the non-phosphorylated form of the target, while phosphorylated Nephrin remained largely constant (Supplementary Fig. 8).

Together, these data on the dynamics of protein amounts and accompanying proteinuria indicated target-specific time-courses as a result of the markedly different stability/turnover rates of the SD core components, their distinct impact on the structural integrity of the SD network or their distinct compensation, all in contrast to the previous conclusions from constitutive knockout mice[11,12,45].

### Significance of selected network constituents for filtration

Finally, we sought to probe a selected set of the newly identified constituents of the SD-associated protein networks for their impact on the filtration process.

For these investigations, we made use of the well-established model of the *Drosophila* nephrocyte as a surrogate for podocytes where functional analysis can be straightforwardly combined with gene-targeting via RNAi-mediated knock-down[46]. In the storage kidney of *Drosophila melanogaster* nephrocytes form SD-homologous junctions which, in a filtration-like process, control the entrance of haemolymphatic contents to lacunae where the resulting filtrate is finally endocytosed (Fig. 4A, C). In this system, FITC-labeled albumin or RFP-tagged ANP are effectively taken up and could thus serve as readily traceable marker substances to quantify the uptake/filtration properties of nephrocytes (Methods[47]). The selection of SD network constituents to be analyzed was based on interactors being not reported before, showing documented expression (of orthologues) in flies, and displaying interesting links in databases and literature.

In wildtype flies, the uptake of FITC and RFP was highly effective, while it was largely decreased or even abolished in flies where expression of the *Drosophila* orthologue of Neph1, Kirre, or of Nephrin, SNS, had been knocked down by RNAi (Fig. 4B, D). Interestingly, knocking down the orthologues of the non-slit-spanning constituents MERTK, ANPRC, MAGI1, ITM2B, or its known interaction partner protein A4[23] also resulted in a significant decrease of the FITC/RFP-uptake,

albeit to distinct degrees (Fig. 4B, D). While, the removal of MERTK and ANPRC reduced the FITC-uptake by about 35% and 20%, respectively, the impairment induced by ITM2B and A4 knock-down was more than 55% of the effect obtained for Kirre with FITC or RFP (Fig. 4B, C). In contrast, knock-down of the podocyte-proteins DLG1[48], RHPN1, CLIC4, ARP3, RPN1, or RHG23, which were not part of the SD-associated network, left the uptake of the fluorescence markers unaltered with respect to WT flies (Fig. 4B, D).

These results indicated, that proper filtration in the fly kidney requires additional constituents to the split-spanning Neph1 and Nephrin protein(s) and suggested important contribution(s) of the previously unknown SD network to the operation of the renal filtration in mammals.

## Discussion

In this work, we show that the SD of native podocytes is a multi-component system that is made up of the slit-spanning Nephrin/Neph1 proteins and a co-assembled high-molecular-weight protein network comprising Podocin and a number of previously unknown transmembrane and cytoplasmic constituents. These previously unknown SD components cover a variety of primary functions predominantly involved in signaling and formation of cellular junctions and are required for the proper operation of filtration in the kidney.

For rigorous unbiased analysis of the protein building blocks of the SD in the rodent kidney, we applied our established proteomic approach that combines multi-AB-based affinity-isolation from the appropriately solubilized source material with high-resolution quantitative mass spectrometry[20–24]. This approach promotes comprehensiveness and specificity of the pursued target interactomes (or proteomes) by means of its key features: (1) use of solubilization conditions that preserve high-molecular-weight complexes of the target (based on careful buffer assessment, Fig. 1C, Supplementary Fig. 3), (2) application of multiple verified ABs with distinct epitopes on each target (Supplementary Fig. 2), (3) MS-based protein quantification with a linear range of about four orders of magnitude and

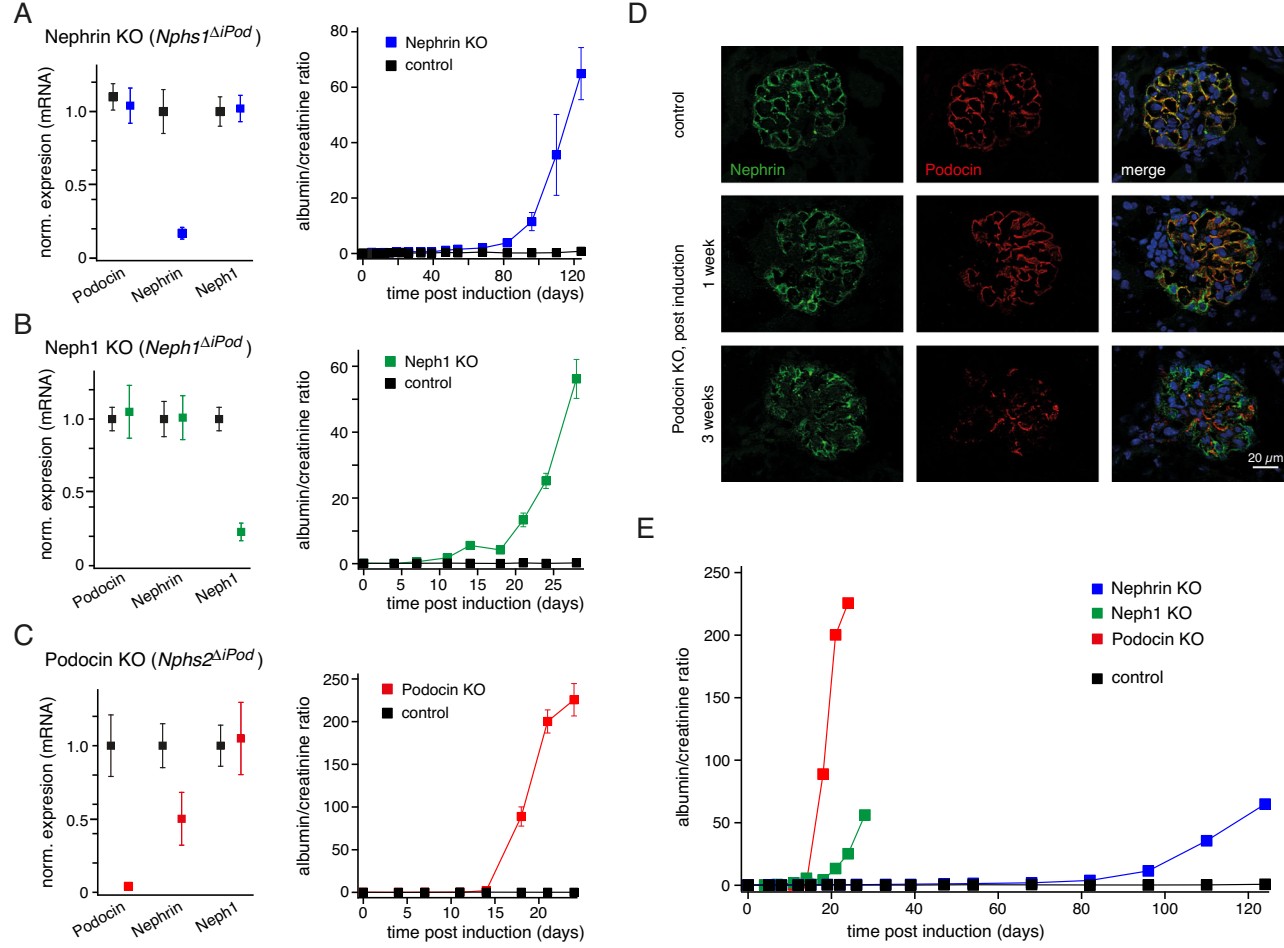

**Fig. 3 | Distinct time-courses of protein stability and development of albuminuria observed after induced knockout of SD core-constituents.**
**A**–**C** Quantitative analysis of mRNAs coding for Nephrin, Neph1, and Podocin three weeks after induction of the target-gene knockout (left panel) and of the albumin/creatinine ratio determined post-induction (right panel) in control and knockout animals. Data are mean (±SEM) of 5 (WT, Nephrin KO), 4 (WT, Neph1 KO), 7 (WT, Podocin KO) (mRNA), and 10, 10, 9, and 6 WT, Nephrin KO, Neph1 KO, and Podocin KO animals (albumin/creatinine ratio), respectively. **D** Immunofluorescence was

determined with anti-Nephrin (green) and anti-Podocin (red) ABs before and at the indicated periods after doxycycline-triggered knockout of Podocin in mouse glomeruli. Blue color originates from DAPI staining. Scaling as indicated; experiments were repeated three times with similar results. **E** Albuminuria quantified as albumin/creatinine ratio as a function of time after induction of the target-gene knockout. Data points are the mean values shown in (A)-(C). Note the largely different development of albuminuria after induction of the targeted knockout.

unbiased identification of all proteins and (4) the use of rigorous negative controls (knockout input and target-unrelated ABs) and (5) consistency among ABs for determination of interactome constituents. In view of the latter criteria, it must be noted that this approach aims at a minimal number of false-positive interactors, deliberately operating at the cost of false negatives.

The interactome of the SD determined with this procedure from two distinct angles, the transmembrane Nephrin/Neph1, and the cytoplasmic membrane-anchored Podocin, confirmed the current view of SD core constituents (Fig. 5A, upper panel) by robust retrieval of both slit-spanning proteins in anti-Podocin APs (Fig. 2). Vice versa, Podocin was abundantly detected in all anti-Nephrin/Neph1 APs, but failed the specificity threshold due to the combined effects of its large abundance and its 'sticky' (multi-valent) binding characteristics to protein-loaded AB-matrices.

In addition, the interactome unveiled the SD as a macro-molecular structure of as yet unknown complexity. Our analyses identified an ensemble of about 25 additional proteins, mostly membrane-associated or transmembrane proteins of different classes, that co-assemble with the core components and thus reconstitute a network-like arrangement of high molecular mass (Fig. 5A, lower panel). Some of the network constituents have been described before as interactors

of Nephrin/Neph1 such as tyrosine kinase Fyn, Dendrin, and ROBO2[29,30,33], while the majority have not been related to the SD or renal filtration. Noteworthy, a number of proteins previously reported to interact with the SD core constituents including TRPC6 ion channels, proteins NCK1, JAM-A (junctional adhesion molecule A), CD2AP or ZO-1 (tight-junction/zonula-occludens protein 1) were either not or not consistently identified by our high-resolution quantitative MS-analyses in anti-Nephrin/Neph1 and/or anti-Podocin APs. Given the sensitivity of at least some interactors to solubilization conditions (Fig, 1D), however, our approach may have failed to detect partner proteins whose binding properties are interfered with by the CL-buffers used. Additional caution is that most of the commercially available ABs partially used in previous work either lacked target-specificity or effective target binding (Supplementary Fig. 2).

Structurally, most of the SD constituent identified here are single-span membrane proteins with rather extended domains protruding to either side of the membrane. Based on current knowledge, the extracellularly located domains comprise sites for ligand/agonist binding, as well as for protein-protein interactions such as the Ig-like or fibronectin domains present in PTPRO, MERTK, BCAM, or ROBO2. The cytoplasmic domains may exhibit enzymatic activities and/or serve scaffolding function(s) (Table 1). Together with the core domains,

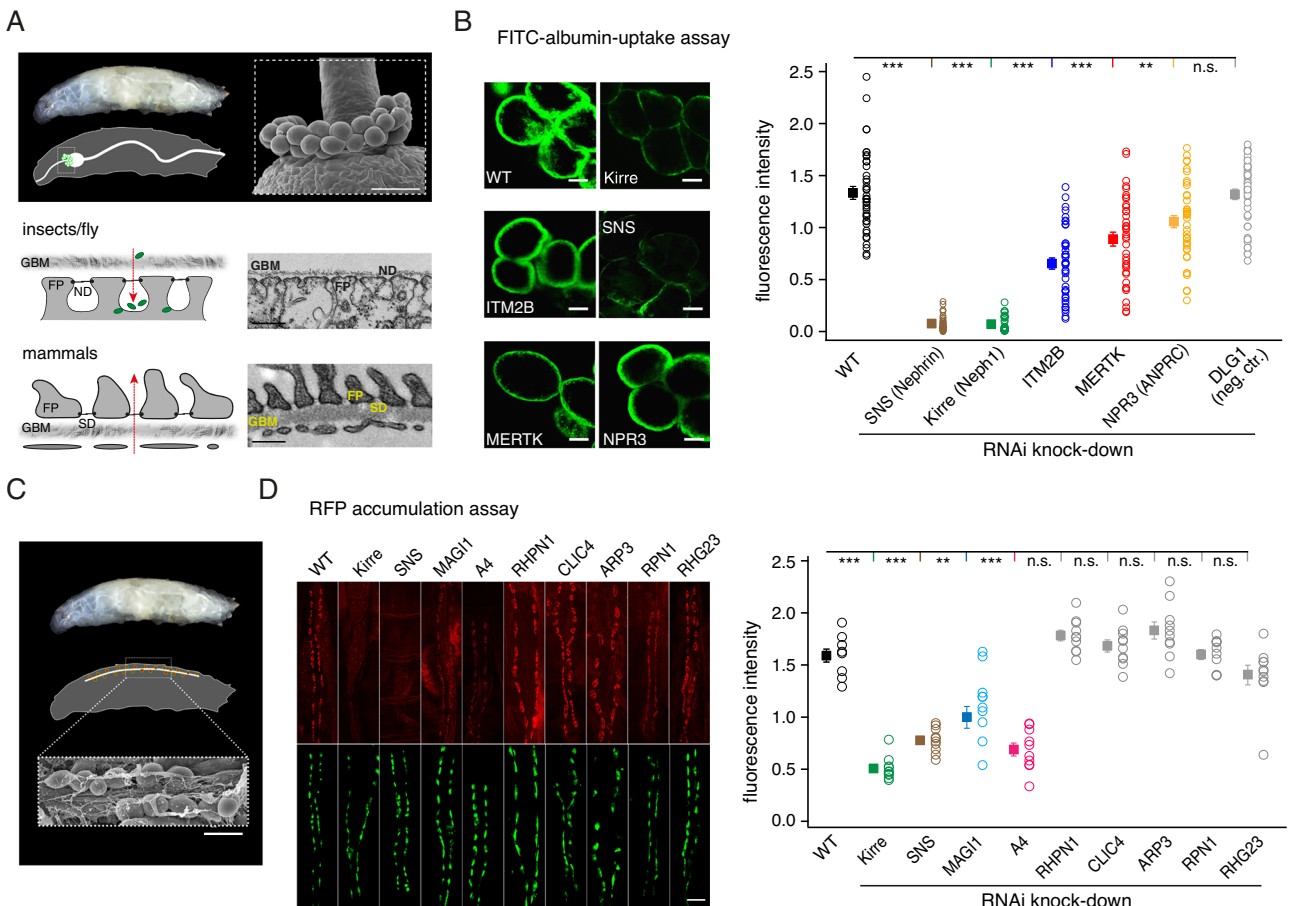

**Fig. 4 | Impaired filtration resulting from knock-down of the identified SD constituents MERTK, ITM2B, A4, MAGI1, and ANPRC in Drosophila nephrocytes. A** Analogy of the renal filter in flies (Garland cell nephrocytes, upper, middle panel) and mammals (podocytes, lower panel) is depicted by EM sections and schemes; red arrows denote the filtration pathways. Upper panel: scanning EM of nephrocytes from fly larvae. Scale bar is 40 μm, result is representative of at least 10 experiments. Note the conserved building of the SD in both species, reflected by FP, GBM, and S/ND (scale bars are 250 nm). **B** Left panel: Representative fluorescence images obtained from nephrocytes without (WT, control) and with RNAi-mediated knock-down of the indicated proteins in the FITC-uptake assay. Scale bars are 10 μm. Right panel: Results of the FITC-uptake assay performed in flies with and without protein knock-down. Data points are results from individual flies, squares are mean ± SEM of 40 (WT), 33 (KIRRE), 38 (SNS), 39 (ITM2B), 38 (MERTK), 40 (NPR3), and 36 (DLG1) experiments. Values for KIRRE, SNS, ITM2B, MERTK, and NPR3 were significantly different from WT with $p < 0.01$ (**) and $p < 0.001$ (***), two-sided Mann–Whitney $U$ test), value for DLG1 knockdown was not significantly different from WT (n.s.). Note the distinct impairment of FITC-uptake determined

with knockdown of MERTK, ITM2B, and NPR3 compared to knockdown of KIRRE and SNS (maximal effect). **C** Scheme and scanning EM depicting pericardial nephrocytes used for the RFP-accumulation assay. Scale bars are 50 μm, and the result is representative of at least 10 experiments. **D** Left panel: representative fluorescence images obtained from individual larvae without (WT, control) and with RNAi-mediated knock-down of the indicated proteins. eGFP marks (lower panel) the individual pericardial nephrocytes, while the RFP signal (upper panel) denotes the uptake of RFP-tagged ANP from the hemolymph into these cells. Scale bars are 50 μm. Right panel: Results of the continuous RFP-accumulation assay performed in flies with and without protein knock-down. Data points are results from individual larvae, squares are mean ± SEM of 8 (WT) and 10 (all others) experiments. Values for KIRRE, SNS, A4, and MAGI1 were significantly different from WT with $p < 0.01$ (**) and $p < 0.001$ (***), two-sided Mann–Whitney $U$-test), values for all other knock-downs were not significantly different from WT (n.s.). Note the distinct impairment of RFP-accumulation determined with knock-down of MAGI1 and A4 compared to knockdown of KIRRE and SNS (maximal effect).

these novel SD building blocks may be envisaged to form a junctional layer between neighboring FPs that is anchored on either side in an extended network of proteins (Fig. 5A, lower panel). In this framework, the network will serve a dual role: First, it provides, via several constituents, anchoring sites for Nephrin and/or Neph1 at the opposite FP, and, second, it tightly links the extracellular layer to both the cytoskeleton and the cytoplasmic signaling environment on the same FP[7] (Fig. 5A, lower panel). Such arrangement is reminiscent of the proteinaceous bridges formed in CNS synapses by cleft-spanning single transmembrane-domain proteins such as Neurexins and Neuroligins and their associated networks on the pre- and postsynaptic membranes (reviewed in ref. 49). Interestingly enough, several of the SD constituents (including PTPRO, MAGIs, and A4-ITM2B-complexes) are also found in the synaptic compartment where they are involved in the formation, maintenance, and signal transduction of synapses[23,34–38].

A few further aspects related to the structure of the SD network and its stability were provided by the quantitative proteomic analysis of the remodeling occurring in the SD upon induced knockdown of its core components Nephrin, Neph1, and Podocin (Fig. 3, Supplementary Figs. 7, 8). Thus, the presence (or stability) of proteins Dendrin, MAGI2, PTPRO, and FYN relies on the presence of the three core components, pointing to direct interactions between these network constituents (Supplementary Fig. 8). Strikingly, the stability of Nephrin was controlled by the phosphorylation status of an acidic cluster immediately adjacent to its transmembrane domain. While the dephospho-form readily decreased after induced knockout, the phospho-form (modified at three sites simultaneously) essentially remained unaffected (Supplementary Fig. 8) suggesting higher stability due to its integration into the SD network and/or counter-regulatory activity of kinase(s). Whether or not such phospho-protection may also occur with Neph1 has to

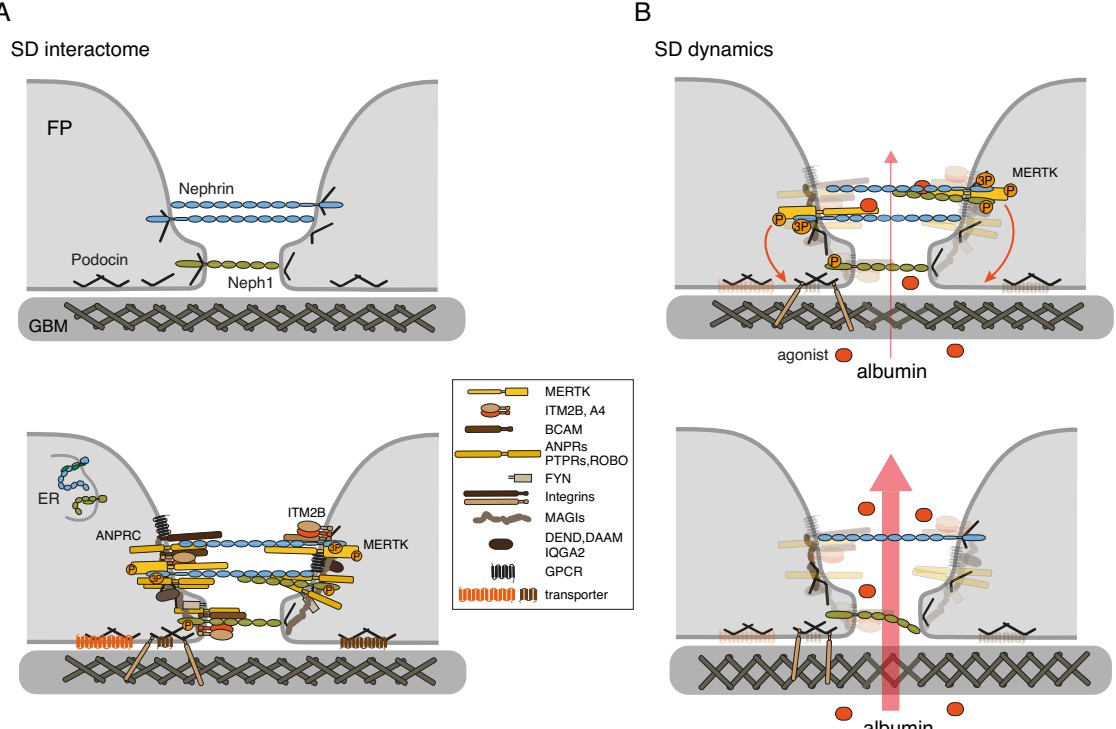

**Fig. 5 | Complexity of SD subunit composition and principle of context-dependent dynamics endowed by the associated network of proteins.**
**A** Schematic representation of the hypothetical molecular appearance of the rodent SD before (upper panel; Grahammer et al.[7]) and after interactome analysis (lower panel). Protein constituents are drawn about to scale (using data from literature and databases, as well as from AlphaFold[59]), and illustration was restricted to structurally distinct constituents (legend on the right) to highlight the complexity of the SD and the associated protein network. Sites and order of interaction, as well as structural details of individual network constituents were not intended. **B** Hypothetical model for context-dependent dynamics introduced into the SD via the associated network of proteins exemplified by the operation and signaling of kinase Mer (MERTK). Briefly, ligand-triggered enzymatic activity may initiate intracellular signaling pathways or direct/indirect structural arrangement(s) required for proper operation of the filtration process via the GBM. Alteration or ablation of this enzymatic activity will lead to increased permeability of the filter finally leading to proteinuria. For further details see text.

remain open, as respective peptides for MS analysis were not available. Nonetheless, the slit-spanning Neph1 appears to be of particular importance for the structural stability of the SD network. Its decrease correlated with the rising phase of albuminuria in all SD core-KOs, even in Nephrin KO mice, where albuminuria was only observed 12 weeks after KO induction together with a robust decrease in Neph1 amounts (Supplementary Fig. 7A). Thus, Neph1 may temporarily compensate for the decrease in Nephrin, while Nephrin was unable to provide compensation for Neph1 (Supplementary Fig. 7B, Supplementary Data 3).

Finally, it should be emphasized that the interactome determined here for Podocin indicated that this protein may in fact serve diverse functions. In addition to its established role as an anchor of the SD, it may act as a cytoplasmic interactor/linker for a number of SLC-type and other transporters in the plasma membrane of podocytes (Fig. 2D, Table 2). As a consequence, Podocin did not exclusively localize to the SD in freeze-fracture replicas, but rather displayed a broader distribution over the plasma membrane of the podocytes (Fig. 2B).

First evidence for the functional relevance of the SD-associated protein network comes from the impaired filtration process induced in fly nephrocytes by the knock-down of four distinct network constituents (Fig. 4). These constituents were selected for their novelty (not associated with the SD before) and for their representation of the distinct classes found in the SD network: Components with enzymatic activity (MERTK), receptor-triggered signaling (ANPRC) and scaffolding function (ITM2B and its interacting partner A4[23]). Albeit differing in their efficiency, these SD constituents identified here all contributed to the proper operation of the filtration process, although they are not part of the slit-spanning layer per se. Noteworthy, impaired filtration was selectively observed with the knock-down of identified

constituents of the SD network, but not with a series of other established podocyte proteins expressed in fly nephrocytes (Fig. 4). Despite identification in rodents, the orthologues of the SD constituents proved to be important for nephrocyte-based filtration in Drosophila.

Based on these observations, and on the dependence of the SD network on the phosphorylation status of Nephrin (Supplementary Fig. 8) it seems well justified to reflect on the molecular role of the complex SD for filtration, although most of the protein components identified here have not yet been mentioned in literature with respect to renal filtration and detailed functional analyses in podocytes are currently missing. Accordingly, we propose a hypothetical working model where the SD-associated network endows renal filtration with context-dependent dynamics through individual or combined actions of its constituents (Fig. 5B).

Such dynamic behavior is exemplified for MERTK, a member of the TAM-family of (tyrosine) kinases which are activated upon agonist/ligand-binding thus switching-on intracellular signaling pathways (including the phosphoinositide kinase 3 pathway (PI3K[50–52]) or mediating posttranslational modification(s) of SD components. These reactions or structural rearrangements may finally converge onto the podocyte cytoskeleton and its anchoring in/to the GBM where they control the filamentous arrangement and permeation properties of the GBM (Fig. 5B) similar to what was recently suggested for a 'gel compression' model[1,26]. Alteration or removal of this signaling will result in defective filtration as observed with the MERTK knock-down (Fig. 4B) and thus eventually lead to proteinuria similarly as observed for reduction/ablation of the SD core components (Fig. 3). Defective function(s) as further hints for the relevance of the identified interactome constituents come from first genetic analyses (Supplementary

Fig. 9). The respective results unraveled profound enrichment of the Podocin interactome constituents among the genes that are either causative for monogenetic glomerular kidney diseases, or that are responsible for abnormal kidney morphology and physiology, as well as albuminuria in genetically manipulated mice (Supplementary Fig. 9).

Given that the vast majority of the identified SD constituents are either involved in cellular signaling or cell adhesion (by the formation of a cellular junction(s)) it is well conceivable that the podocyte SD operates as a molecular sensor and effector which renders renal filtration dynamic and able to adapt to changing conditions. The molecular details behind such context-initiated dynamics, their exact sites of action as well as their reaction target(s) within or adjacent to the SD must currently remain open and should fuel future research.

## Methods

### Animals

All animal experiments were conducted according to the NIH Guide for the care and use of Laboratory animals as well as in accordance with the German law for the welfare of animals, and were approved by local authorities (Regierungspräsidium Freiburg G-14/22). Mice were housed in an SPF facility with free access to chow and water and a 12 h day/night cycle. Breeding and genotyping were done according to standard procedures. *Nphs1* (kind gift from Alessia Fornoni, Miller University, Miami, USA[53]), *Nphs2* (kind gift from Géraldine Mollet and Corinne Antignac, Imagine Institute of Genetic Diseases, Paris, France[54]), and *Neph1*[7] floxed mice on a C57Bl6/Crl background, were crossed with Tg(Nphs1-rtTA*3 G)[8Jhm] (kind gift from Jeff Miner, Washington University, St. Louis, USA) and Tg(tetO-cre)[1Jaw] yielding *Nphs1ΔiPod*, *Nphs2ΔiPod*, and *Neph1ΔiPod* respectively. Doxycyclin (2 mg/ml in 5% sugar solution, Fagron, Barsbüttel, Germany) was administered for 8 days in the drinking water at an age of 5 weeks. Urinary albumin and creatinine were measured using a fluorimetric albumin test kit (Progen, PR2005, Heidelberg, Germany) and enzymatic colorimetric creatinine kit (LT-SYS, Lehmann, Berlin, Germany) following the manufacturer's instructions.

### Isolation of glomeruli

Mouse kidneys were dissected together with the abdominal aorta and transferred into dishes filled with 37 °C prewarmed Hank's buffered salt solution (HBSS). Each kidney was perfused slowly through the renal artery with 2 ml 37 °C bead solution as described previously[55]. Kidneys were minced into 1 mm³ pieces on an ice-cooled petri dish using a scalpel. Minced tissue from one kidney was transferred into a Precellys Tube 2.8 mm (Peqlab, Germany), 1 ml cold HBSS was added and the tube was put into a Minilys apparatus (Peqlab, Germany) for 15 sec at 5000 rpm. The kidney fragments were gently pressed through a 100 μm cell strainer into a 50 ml Falcon tube using ice-cold HBSS. After spin down, the supernatant was discarded and the pellet resuspended in 2 ml cold HBSS. These tubes were inserted into a magnetic particle concentrator and the separated glomeruli were washed twice using cold HBSS. After the removal of HBSS, the tubes were weighed and snap-frozen in liquid nitrogen.

Eight weeks old male CD-1 rats were purchased from Charles River, Germany. Rat Kidneys were removed in deep anesthesia (Ketamine 100 mg/kg and Xylazine 5 mg/kg in 10 μl/g 0.9% NaCl) and immediately perfused with ice-cold HBSS solution via the renal arteries. The renal cortex was removed and minced, and glomeruli were isolated using the sieving method. The following sieve sizes were used: 180 μm, 106 μm, and 53 μm. The tissues were constantly rinsed with 4 °C cold HBSS solution during glomeruli isolation. Glomeruli were collected on the lowest sieve. The purity of renal glomeruli (>95%) was verified using light microscopy. The final glomerular pellet was obtained by centrifugation with 2300 rpm for 7 min at 4 °C and was snap-frozen in liquid nitrogen after weighing.

### Immunofluorescence

For immunofluorescence kidneys were frozen in OCT compound and sectioned at 5 μm (Leica Kryostat, Wetzlar, Germany). The sections were fixed with 4% paraformaldehyde in phosphate-buffered saline (PBS), blocked in PBS containing 5% BSA + 5% Normal Donkey Serum (Jackson Immuno Research, Suffolk, UK), and incubated for 45 min with primary antibodies as indicated. After several PBS rinses, fluorophore-conjugated secondary antibodies (Life Technologies, Darmstadt, Germany) were applied for 30 min. Images were taken using a Zeiss fluorescence microscope equipped with a ×20 and ×63 water immersion objective (Zeiss, Oberkochen, Germany). The following antibodies were used: guinea pig anti-Nephrin (GP-N2; Progen, Heidelberg, Germany), rabbit anti-Podocin (P0372; Sigma, Schnelldorf, Germany), and rabbit anti-Nephrin[7].

### RNA Isolation and qPCR

RNA was isolated using the acidic GTC-Phenol-Chloroform method as previously described[55]. RNA purity was assessed with the help of the 260/280 nm ratio and only ratios between 1.8 and 2.0 were accepted. After DNase digest, the RNA was reverse transcribed using the iScript Reverse Transcription Supermix (BioRad, California, USA) following instructions of the manufacturer. qPCR was performed in a CFX Connect Real-time PCR Detection System (BioRad, California, USA) using SYBR Green as a fluorophore. Each sample was measured in triplicate and normalized to a reference housekeeping gene using the ΔΔC_T method in the CFX Manager Software 3.1 (BioRad, California, USA).

### Primers for qPCR

| Primer | Sequence (5' → 3') |
| --- | --- |
| GAPDH, Unique Assay ID: qRnoCID0057018, Rat | (Bio-Rad Lab., Hercules, California, USA) |
| KIRREL, Unique Assay ID: qRnoCED0052567, Rat | (Bio-Rad Lab., Hercules, California, USA) |
| NPHS1, Unique Assay ID: qRnoCED0017962, Rat | (Bio-Rad Lab., Hercules, California, USA) |
| HPRT forward primer, Mouse | GCTTTCCTTGGTCAAGCAGTACAG |
| HPRT reverse primer, Mouse | GAAGTGCTCATTATAGTCAAGGGCATATCC |
| NPHS1, Unique Assay ID: qMmuCED0045387, Mouse | (Bio-Rad Lab., Hercules, California, USA) |
| NPHS2 forward primer, Mouse | CACAAGGTTGATCTCCGTCTC |
| NPHS2 reverse primer, Mouse | GCAGACAGCGTCTATCTCCATT |
| RPL13 forward primer, Mouse | CTCATCCTGTTCCCCAGGAA |
| RPL13 reverse primer, Mouse | GGGTGGCCAGCTTAAGTTCTT |

### Biochemistry and cell biology

**Membrane isolation from rodent glomeruli.** Isolated rat or mouse glomeruli were resuspended in homogenization buffer (0.32 M sucrose, 10 mM Tris/HCl, 1.5 mM MgCl2, 1 mM EGTA, pH 7.5) supplemented with protease inhibitors (Aprotinin, Pepstatin, Leupeptin, each at 2 μg/ml), 1 mM IAA and 2 mM PMSF; 6 ml of buffer per -1 g of wet glomeruli (~2.5 ml/rat) and homogenized with Dounce homogenizer (20 strokes of tight-fitting pestle). The homogenate was centrifuged at 1050×*g* for 5 min and the supernatant (S1) was collected. Pellets were further homogenized by sonication in fresh homogenization buffer (~2 g per g of starting tissue), centrifuged at 1050×*g* for 5 min, supernatant was collected and centrifuged again at 1050×*g* for 5 min (S1´). For pellets from mouse glomeruli, this procedure was repeated.

Supernatants (S1 + S1´) were combined and centrifuged at 200,000×*g* for 30 min (T865 or T647.5 rotor, Sorvall) or for 25 min

(S80-AT3 rotor, Sorvall). The resulting pellet was suspended in hypotonic lysis buffer (5 mM Tris/HCl, pH 7.4, 1 mM EDTA) supplemented with protease inhibitors (at 2 µg/ml), 2 mM IAA, and 2 mM PMSF (4–5 ml per g of starting tissue), incubated on ice for 30 min with gentle agitation and subsequently centrifuged at 200,000×g for 30 min (T865 or T647.5 rotor, Sorvall) or for 25 min (S80-AT3 rotor, Sorvall). The resulting pellet was suspended in 15 ml of 0.5 M sucrose (0.5 M sucrose in 10 mM Tris/HCl pH 7.5, 1 mM EDTA) (-2 ml per g of starting tissue), underlaid with 1.3 M and 1.98 M sucrose and centrifuged at 135,000×g for 60 min (Surespin630/36 or SW30 rotor, Sorvall) or for 45 min (S52-ST rotor, Sorvall). The interface between 0.5 M and 1.3 M sucrose layer was collected with a syringe, diluted with 20 mM Tris/HCl pH 7.5, and centrifuged at 200,000×g for 20 min (T865 or S80-AT3 rotor, Sorvall). The resulting pellet was suspended 20 mM Tris/HCl pH 7.5 to obtain the membrane fraction ('MF', Supplementary Fig. 1) used as source material for APs.

**Affinity-purification from membrane fractions.** After determination of the MF protein concentration by a Bradford assay, 0.4 mg (rat) or 0.1 mg (mouse) of MF were solubilized with CL-47 or CL-91 (Logopharm GmbH, Germany) for 30 min on ice (at 1 mg protein/ml). After clearing by ultracentrifugation (10 min, 100,000×g (rat) or 9 min, 135,000×g (mouse)) solubilisates were incubated with antibodies, 15 µg (rat) or 7.5 µg (mouse) (Supplementary Fig. 2) immobilized on magnetic beads (Invitrogen, 10002D or 10004D). After two hours of incubation and two brief washes, proteins were eluted with 1× Laemmli buffer without DTT, shortly run on 10% SDS–PAGE gels, and silver-stained. Lanes were cut into two sections (high (->65 kDa) and low MW (-< 65 kDa)) and digested with sequencing-grade modified trypsin (Promega, Mannheim, Germany). Peptides were extracted and prepared for MS analysis as described[22].

**Membrane fraction preparation for MS.** Membrane fractions isolated from rat/mouse glomeruli were run on 10% SDS-PAGE gel and silver-stained. Lanes were cut into four sections (>150 kDa, >75 kDa, >37 kDa, and <37 kDa) or three sections (>100 kDa, >37 kDa, and <37 kDa) (Supplementary Fig. 7) and digested with sequencing-grade modified trypsin (Promega, Mannheim, Germany). Peptides were extracted and prepared for MS analysis as described[22].

**Protein assembly analysis by BN-PAGE.** Protein complexes were separated by BN-PAGE essentially as described[22]. Solubilisates (CL-47 or CL-91) of membrane fraction from rat glomeruli (Fig. 1C, S3), were mixed with 0.05% Coomassie G-250 and loaded on linear gradient gels (3–12% polyacrylamide). Gels were run with 15 mM BisTris/50 mM Tricine/0.02% Coomassie G250 as running buffer and 50 mM BisTris (pH 7.0) as anode buffer. Mitochondrial membrane protein complexes were used as a molecular weight standard for the estimation of complex sizes in the first dimension. BN-PAGE lanes were excised and denatured in SDS buffer and subsequently placed on top of 8% SDS-PAGE gels.

**Immunoblot detection.** SDS-PAGE resolved proteins were transferred on PVDF-membranes and detected with target-specific antibodies: anti-Nephrin (Progen #GP-N2, lot 504271, 1:1000), anti-Neph1[7] (1:250–1:1000), anti-Podocin (Sigma #P0372, lot 035M4851V, 1:1000–2000), anti-ITM2B (Santa-Cruz #sc-50026, lot A2407, 1:200), anti-ATP1A1 (Santa-Cruz #sc-21712, lot L3013, 1:200), anti-HRP-conjugated secondary ABs (Santa-Cruz: sc-2004 (goat anti-rabbit IgG-HRP, lot H1015, 1:25000; sc-2903 (goat anti-guinea pig IgG-HRP, lot I2107, J0812), 1:10,000-1:50,000; sc-2005 (goat anti-mouse IgG-HRP, lot H2014), 1:25000; sc-2768 (rabbit anti-goat IgG-HRP, lot J0713), 1:50,000; Abcam: ab7090 (Goat Anti-Rabbit IgG H&L (HRP) pre-adsorbed, lot GR270768-25), 1:10,000; or Cell Signalling: 7074 (goat Anti-rabbit IgG, HRP-linked Antibody, lot 28), 1:2000) in combination with ECL Prime (GE Healthcare, Germany) were used for visualization.

**Antibody generation.** Rabbit polyclonal antibody against Neph1 was generated by Pineda Antikörper (Berlin, Germany)[7]. All other rabbit polyclonal antibodies against Nephrin, Neph1, and Podocin were generated by AbFrontier (Seoul, Korea). Immunizing peptides are listed in Supplementary Fig. 2. Podocin fragment used to generate the C-term antibody was based on Roselli et al.[13].

**Mass spectrometry**

**LC-MS/MS analysis.** Mass spectrometric analyses of tryptic peptide mixtures were carried out on an Orbitrap Elite or Orbitrap Q Exactive HF-X (Supplementary Fig. 7) hybrid mass spectrometer (Xcalibur 2.2 sp1) coupled to an UltiMate 3000 RSLCnano HPLC system (all Thermo Fisher) as described[56]. Specifically adapted parameters were (i) for AP-MS analyses (Figs. 1D, 2D): aqueous organic LC gradient (eluent 'A': 0.5% acetic acid; eluent 'B' 0.5% acetic acid in 80% acetonitrile; gradient: 5 min 3% B, 60 min from 3% B to 30% B, 15 min from 30% B to 99% B, 5 min 99% B, 5 min from 99% B to 3% B, 15 min 3% B), MS2 maximum injection time = 200 ms, dynamic exclusion duration = 30 s, minimum signal threshold for fragmentation = 2000 counts, (ii) for MS quantification of total membrane fraction (Supplementary Fig. 1B): LC gradient (5 min 3% B, 120 min from 3% B to 30% B, 20 min from 30% B to 99% B, 5 min 99% B, 5 min from 99% B to 3% B, 15 min 3% B), MS2 maximum injection time = 200 ms, dynamic exclusion duration = 60 s, minimum signal threshold for fragmentation = 5000 counts, (iii) for MS quantification of total membrane fractions (Supplementary Fig. 7): LC gradient (1 min 3% B, 120 min from 3% B to 30% B, 20 min from 30% B to 40% 'B', 10 min from 40% B to 50% B, 5 min from 50% B to 99% B, 5 min 99% B, 5 min from 99% B to 3% B, 10 min 3% B, MS2 maximum injection time = 100 ms, dynamic exclusion duration = 60 s, intensity threshold for fragmentation = 40,000 counts.

**Protein identification.** LC-MS/MS data were extracted using "msconvert.exe" (part of ProteoWizard; http://proteowizard.sourceforge.net/, version 3.0.6906). Peak lists were searched against the UniProtKB/Swiss-Prot database (rat, mouse, and human entries; release 2019_11, 2022_01) using Mascot 2.6. (Matrix Science, UK). Initially, preliminary searches with high peptide mass tolerance (±50 ppm) were performed and after linear shift mass recalibration using in-house developed software, mass tolerance was reduced to ±5 ppm for final searches. Fragment mass tolerance was set to ±0.8 Da (ion trap MS/MS spectra). One missed trypsin cleavage and common variable modifications including S/T/Y phosphorylation were accepted for peptide identification. The peptide significance threshold was set to $p < 0.05$. Proteins identified by only one specific MS/MS spectrum or representing exogenous contaminations such as keratins or immunoglobulins were eliminated.

**Protein quantification.** Label-free quantification of proteins was carried out using a recently developed procedure[25,57]. Briefly, peptide signal intensities (peak volumes, PVs) from FT full scans were extracted and offline mass calibrated using MaxQuant v1.6.3 (http://www.maxquant.org). Then, peptide PV elution times were pairwise aligned using LOESS regression (reference times dynamically calculated from the median peptide elution times over all aligned datasets). PVs were then assigned to peptides based on congruent m/z and elution times (±1 min/2–3 ppm) as obtained directly or indirectly from MS/MS-based identification using in-house developed software as described[16].

In a second step, protein-specific PV data matrices (peptides versus MS runs) were subjected to analysis of their internal consistency (between runs and with regard to their relative peptide ionization efficiencies). From these relations between PV values, expected PV values were calculated to derive a measure of accuracy for each PV value that was subsequently used as a weighting factor. In parallel, time- and run-dependent detectability thresholds were estimated for each measured or missed PV value in the matrix. Global protein references were then obtained from the aggregation of qualified PV

data in each protein matrix, termed protein reference ridges (basically vectors reflecting the maximum protein coverage of MS/MS-identified and quantified peptides with their relative ionization efficiencies). Protein quantification was finally achieved by weighted fitting of its measured peptide PVs to its reference ridge. In case no (consistent) peptide PVs could be identified, an apparent protein detection limit was determined using the detectability thresholds of the three best ionizing peptides fitted to the protein reference ridge. Molecular abundances were calculated as abundance$_{norm}$spec values[16] from the fitted protein ridges. The molecular abundances determined by this procedure are basically comparable to MaxQuant LFQ values (where all data is fitted without weighting) but more accurate, in particular for proteins quantified from sparse PV data.

To determine specificity in affinity purifications (APs), ratios of molecular abundances of proteins in WT versus control APs were calculated, logarithmized (basis 10) and normalized to the abundance ratio of the respective target (=maximum 100%) and to the estimated distribution of the background proteins (=minimum 0%) as shown in Supplementary Fig. 5. The resulting target-normalized ratios and molecular abundances were collectively inspected using the BELKI software suite (Jordan, J, and Haupt, A (2021). Belki: Version 2.0 (v2.0), Zenodo. https://doi.org/10.5281/zenodo.4670785). For anti-Nephrin/ Neph1 APs (Fig. 1) three target-unrelated ABs were used for control APs (rabbit IgG (Millipore, 12–370, lot 2295402); anti-PODXL (Abcam, ab205350, lot GR268901–3); anti-SLC34A3 (Biorbyt, orb313245, lot BS6202)). Visualization of tnR-values by t-distributed stochastic neighbor embedding (tSNE, Fig. 2C) suggested a minimum tnR of 0.20 as indicator for positive interaction. For anti-Podocin APs (Fig. 2), APs from membrane fractions prepared from Podocin KO animals with additional depletion of the small remainders of the target protein via the a1 anti-Podocin AB were used as controls and a minimum tnR of 0.25 was used as indicator for positive interaction.

## FITC-uptake assay in *Drosophila* nephrocytes

*D. melanogaster* stocks were cultured on standard cornmeal molasses agar food and maintained at 25 °C. The Drosophila homologs of mammalian genes were identified using the DIOPT DRSC Integrative Ortholog Prediction Tool (Harvard Medical School)[58]. RNAi-Based Nephrocyte Functional Screen Procedure: Virgins of prospero-Gal4 (gift from Barry Denholm, University of Edinburgh, Edinburgh, UK), MHC-ANF-RFP, HandGFP, and Dot-Gal4 transgenic lines (gift from Zhe Han, University of Michigan, Ann Arbor, USA) were crossed to UAS-CG3653-RNAi (KIRRE, NEPH1, VDRC 27227/GD); UAS-CG3662–RNAi (ITM2B, Bloomington Drosophila Stock Center 18281); UAS-CG2272-RNAi (MERTK, VDRC 33518/GD); UAS-CG14877-RNAi (NPR3, VDRC 45324/GD); UAS-CG1725-RNAi (DLG1, VDRC 109274/KK) males at 25 °C. Two days after crossing, flies were transferred to small collection cages with grape juice agar plates to collect the embryos for 24 hours at 25 °C. Collected embryos were aged for 48 hours at 29 °C. Nephrocytes of L3 larva were dissected, transferred onto a cover slip, and subjected to a 30 s pulse of FITC labeled albumin. For each experiment 10 dissected larvae were used. Using a confocal microscope (A1 CLEM Eclipse TI./Nikon) equipped with a ×60 oil-immersion objective, fluorescent and DIC images of each nephrocyte package were taken. Blinded to the genotype 4 randomly chosen cells of nephrocyte package were used to quantify the uptake efficiency using the ImageJ Software (NIH, USA). Each condition was repeated twice.

## RFP-accumulation assay in *Drosophila* nephrocysts

*D. melanogaster* stocks were cultured on standard cornmeal molasses agar food and maintained at 25 °C. The Drosophila homologs of mammalian genes were identified using the *DIOPT DRSC Integrative Ortholog Prediction Tool (Harvard Medical School)*. RNAi-Based Nephrocyte Functional Screen Procedure: Virgins from MHC-ANF-RFP, HandGFP, and Dot-Gal4 transgenic lines (gift from Zhe Han,

University of Michigan, Ann Arbor, USA) were crossed to UAS-RNAi male flies (Kirre-RNAi: VDRC109585KK; Sns-RNAi: VDRC109442KK; Magi-RNAi: VDRC41736GD; Vha100-2-RNAi: VDRC109763KK; Rhophilin-RNAi: VDRC110377KK; Clic-RNAi: VDRC105975KK; Arp3-RNAi: VDRC108951KK; CG33303-RNAi: VDRC107778KK; RhoGAP19D-RNAi: VDRC43955GD at 25 °C. Two days after crossing, flies were transferred to small collection cages with grape juice agar plates to collect the embryos for 24 hours at 25 °C. Collected embryos were aged for 48 hours at 29 °C and then subjected to the examination of RFP-accumulation in pericardial nephrocytes using a confocal microscope. The RFP mean fluorescence intensity of GFP positive areas was measured to quantify the uptake efficiency.

## EM, SDS-digested freeze-fracture replica immune-labeling (SDS-FRL)

Pellets of freshly isolated rat glomeruli (as described above) were fixed in 0.1 M phosphate buffer (PB) containing 2% PFA and 15% saturated picric acid for 1 h, then cryoprotected in 0.1 M PB with 30% glycerol overnight (O/N) at 4 °C. Aliquots of glomerular masses were frozen under high-pressure (HPM100, Leica, Austria). The frozen samples were fractured at −140 °C and the fractured facets were coated with carbon (5 nm), platinum-carbon (2 nm), and an additional layer of carbon (18 nm) in one of the freeze-fracture replica machines (ACE 900, Leica, or BAF 060, BAL-TEC, Liechtenstein). Replicas were digested for two O/ Ns at 60 °C followed by one O/N at 37 °C in 15 mM Tris buffer (TB, pH 8.3) containing 2.5% sodium dodecyl sulfate (SDS) and 20% sucrose.

Prior to immunohistochemical labeling, replicas were digested with collagenase D (2 mg/ml, Roche Diagnostics GmbH, Mannheim, Germany) diluted in PBS for 2 h at 37 °C. After rinsing in washing buffer (0.05% bovine serum albumin (BSA), 0.1% Tween-20, made up in 50 mM Tris-buffered saline (TBS)), replicas were blocked in TBS containing 5% BSA and 0.1% Tween-20 (blocking buffer) for 1 h at room temperature (RT). Proteins of the glomerular SD were labeled with one of the following primary antibodies: anti-Podocin a1 (rabbit; 1:100 (10 μg/ml), Sigma-Aldrich, St. Louis, MO) or anti-Nephrin a2 (goat; 1:30 (7 μg/ml), R&D Systems, Minneapolis, MN) made up in 50 mM TBS containing 1% BSA and 0.1% Tween-20 O/N at RT. After thorough washing, replicas were reacted either with 12 nm gold-coupled donkey anti-rabbit IgG (Podocin) or with 12 nm gold-coupled donkey anti-goat IgG (Nephrin) secondary antibodies (1:30, Jackson ImmunoResearch Europe, Cambridgeshire), diluted in blocking buffer, O/N at RT. Replicas were analyzed for subcellular localization of Nephrin and Podocin with an electron microscope (JEM 2100 Plus) as described previously[54].

## Histology and EM

Kidneys were perfusion fixed in 4% phosphate-buffered paraformaldehyde, embedded in paraffin, and further processed for PAS (Periodic Acid–Schiff) staining. For ultrastructural analysis kidneys were also fixed in 4% phosphate-buffered paraformaldehyde. Samples were postfixed in 1% osmium tetroxide in the same buffer for 1 h and stained en bloc in 1% uranyl acetate in 10% ethanol for 1 h, dehydrated in ethanol, and embedded in LX112 (Fisher-Scientific, Schwerte, Germany). Semithin sections were stained with toluidine blue. Thin sections were stained with uranyl acetate and lead citrate and examined in a Jeol JEM 1200EX electron microscope (JEOL, Eching, Germany).

## Statistics and reproducibility

All values were reported as mean ± SEM of entire sets of data (without exclusion of data). Statistical significance was tested using a nonparametric two-tailed Mann–Whitney $U$-test (Fig. 4) or one-sided Fisher's exact test (Supplementary Fig. S9) for analysis of differences in group proportions. EM pictograms, Western-blotted gels, and confocal images shown are representative results of successfully repeated experiments with identical results. Calculations were performed in Microsoft Excel or IGOR Pro 9. Statistical differences with $p < 0.01$ were

considered significant. In Fig. 4, double (**) and triple asterisks (***) indicate $p < 0.01$, and $p < 0.001$, respectively.

## Reporting summary

Further information on research design is available in the Nature Research Reporting Summary linked to this article.

## Data availability

The MS proteomics data generated during this study are available via ProteomeXchange with identifier PXD034356 https://doi.org/10.6019/PXD034356; any other data supporting the findings of this study are available from the corresponding authors upon reasonable request. Source data are provided with this paper.

## Code availability

Parts of the MS evaluation work implied the use of the in-house developed software BELKI: Jordan, J and Haupt, A (2021). Belki: Version 2.0 (v2.0), Zenodo. https://doi.org/10.5281/zenodo.4670785.

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

## Acknowledgements

We thank G. Walz for fruitful discussions and critical reading of the manuscript, J. Jordan for help with the evaluation of meAP-MS data, and Charlotte Meyer, Natalie Wernet, and Valerie Oberüber for excellent technical support. This work was supported by grants from the Deutsche Forschungsgemeinschaft (DFG, German Research Foundation) SFB 1140 (A01, A03 (to B.F., T.B.H., F.G.), SFB 1453 (project-ID 431984000 (to B.F. and A.Kö.), SFB 1192 (B02, B08, B09 (to F.G., T.B.H.), SFB 1381 (project-ID 403222702 (to B.F) FA 332/16-1 (to B.F., F.G.), GR 3933/1-1 (to F.G.), HU 1016/8-2,11-1, 12-1 (to T.B.H.) and FOR 2143 (to A.Ku.).

## Author contributions

F.G., T.B.H., J.S., and B.F. conceived the project, M.K.K., J.S., W.B., and U.S. performed experiments related to protein biochemistry and proteomic analyses, H.A., B.D., S.W., L.K., U.A., C.T., O.K., and F.G. analyzed the knock-out animals, M.H. and P.T. characterized SD function in flies, C.A. and G.M. provided breeding pairs for *Nphs2* knockout mice and plasmids for AB generation, A.Kö. and B.Gö. performed genetic analyses, and A.Ku. and B.Ga. did immuno-EM in SDS-FRL configuration. B.F. and F.G. wrote the manuscript with the support of all authors.

## Funding

## Competing interests

The authors declare no competing interests.

## Additional information

[1]Institute of Physiology, Faculty of Medicine, University of Freiburg, Hermann-Herder-Str. 7, 79104 Freiburg, Germany. [2]III. Department of Medicine, University Medical Center Hamburg-Eppendorf, Hamburg, Germany. [3]Department of Medicine IV, Faculty of Medicine, University of Freiburg, Hugstetterstrasse 49, 79104 Freiburg, Germany. [4]Department of Anatomy, Histology and Embryology, Faculty of Medicine, University of Debrecen, Nagyerdei krt. 98, Debrecen 4032, Hungary. [5]Laboratory of Hereditary Kidney Diseases, Imagine Institute, Paris Descartes University, INSERM U1163, 75015 Paris, France. [6]Institute of Genetic Epidemiology, Faculty of Medicine and Medical Center, University of Freiburg, Hugstetter Str. 49, 79106 Freiburg, Germany. [7]Signalling Research Centres BIOSS and CIBSS, Schänzlestr. 18, 79104 Freiburg, Germany. [8]These authors contributed equally: Maciej K. Kocylowski, Hande Aypek. [9]These authors jointly supervised this work: Tobias B. Huber, Bernd Fakler, Florian Grahammer. ✉e-mail: t.huber@uke.de; bernd.fakler@physiologie.uni-freiburg.de; f.grahammer@uke.de

