## [Peer Review File · Nature Communications]

A slit-diaphragm-associated protein network for dynamic control of renal filtrationREVIEWER COMMENTS

Reviewer #1 (Remarks to the Author):

In this study the authors investigated the network of the molecules of the kidney podocyte slit diaphragm with high-resolution proteomic analysis. Then, the function of the molecules identified are investigated. The study provides some of important findings for better understanding the structure and function of the slit diaphragm. However, there are some concerns for this paper. The followings may help to improve the manuscript.

- (1) The advantages of the techniques adopted in this study should be explained more precisely.
- (2) Some proteins, which are reported to be components of the slit diaphragm, are not detected with the method in this study. That seems to be an important point for assuring the availability of the method in this study. More explanations are needed.
- (3) The findings shown in Figure 4C are not clear enough to discover the function of the molecules. More explanations on the interpretation of the results are needed.

Reviewer #2 (Remarks to the Author):

In this manuscript, the authors discovered new proteins that interact with key components of the Slit Diaphragm (SD) using proteomic analysis on membrane fractions from glomeruli of rat & mouse kidney. Using antibodies against Nephtrin, NEPH1 and Podocin, they identified many (over 30) novel proteins that have not been shown before to be interacting with the SD. Using *Drosophila* nephrocytes, they tested the homologs of three of these proteins (MERTK, ANPRC & ITM2B) and showed that their fly homologs are needed for the filtration function. The manuscript is well-written, and it is significant to identify many potential new proteins that interacts with SD, and validate three of them in the fly nephrocyte. The findings are potentially significant for the podocyte field. However, there are some flaws in the data interpretation and more discussion / explanation are needed.

Major Concerns:

1. The authors set the threshold for specific binding as $\text{tnR} > 0.2$ or 0.25 in Figures 1 and Figure 2. This threshold is critical for claiming that the newly identified proteins from this study are not false positive. The authors have used t-SNE for the separation of true binding vs. non-specific binding. However, the separation doesn't look great in Figure 2C, and it is not clear how the authors determined the $\text{tnR} > 0.2$ or 0.25 cutoff.
2. Podocin was previously established as a trans-membrane protein associated with the lipid raft and directly binds to Nephtrin and CD2AP in lipid rafts (Schwarz et al., JCI 2001). However, in this study, Podocin was described as a cytoplasmic protein, which serves as a "universal cytoplasmic anchor targeting not only the SD, but also integral membrane proteins with diverse functions". This is contradictory with a well-established understanding of our previous knowledge about Podocin and needs to be explained in more details.
3. CD2AP was shown to be directly binding to both Podocin and Nephtrin. However, it was not found among the proteins identified in this study. Why is that?
4. The authors used *Drosophila* nephrocytes to show that the fly homologs of three proteins (MERTK, ANPRC & ITM2B) identified from this study are required for the filtration function, but didn't explain why these three proteins were chosen. Are these proteins the only ones that are conserved in flies? If not, why choose these three instead of others?

5. The nephrocyte RNAi experiments lack sufficient demonstration of RNAi knock-down efficiency. Alternatively, the authors could use independent RNAi lines to validate the knock-down results to avoid any off-target effects.

6. The model in Figure 5A is mainly based on speculation. The data from this study only support that these proteins are in close range of the key SD proteins (Nephrin, NEPH1 and Podocin). It is not clear to this reviewer how these specific domain binding, inside or outside of the membrane, could be suggested by the results from this study.

7. The model in Figure 5B is also mainly based on speculation. If the authors believe that MERTK is the key protein that controls SD dynamics, some data to support this idea should be shown. The author mentioned that "Alteration or removal of this signaling will result in defective filtration as observed with the MERTK knock-down and thus eventually lead to proteinuria (Fig. 3).", but this reviewer could not find any MERTK knock-down data in Figure 3. Is this a typo?

Reviewer #3 (Remarks to the Author):

The manuscript titled "A slit-diaphragm-associated protein network for dynamic control of renal filtration" by Kocylowski and colleagues utilizes rodent kidney tissue extracts and affinity purifications of various known slit-diaphragm-associated proteins. Isolated complexes are then analyzed by LC-MS to generate a protein network of SD-associated protein. Using both genetic mouse models and a drosophila nephrocyte system to evaluate filtration process, functional relevance for several newly identified SD components was demonstrated.

Overall, this is an interesting manuscript that uses rigorous biochemical purification of SD components to establish a slit-diaphragm-associated protein network. This component of the manuscript is certainly novel and without a doubt relevant to the renal community. This part of the paper will have a significant resource value.

A limited number of newly identified proteins are then evaluated in a drosophila functional assay. While this component of the manuscript is significantly briefer, it is appropriate for a more proteomics centric manuscript.

There are a couple of comment the authors should address to improve presentation and clarity of their results.

1) Considering this paper is very proteomics centric the authors need to deposit their raw data to a depository. Most importantly the authors need to provide supplemental tables of their filtered mass spec data. This way others can mine their data. I am very surprised this is missing.

a. Along these lines. Figure 1D/Table 1 and Figure 2D/Table 2 need more information to be useful. In the figures the authors should add MaxQuant LFQ values (it seems this algorithm was used). If proteins weren't detected (or imputed) this needs to be shown in this figure as well (for example in the white boxes – I would assume).

b. Why was IP-MS done in triplicates for figure 1 and only in duplicates for figure 2?

c. In the tables the authors should also add the total number of peptides (or spectra) that were detected for each protein reported.

d. It seems a bit unclear (although somewhat described in the methods) why a tnR ratio of >0.2 was selected in Figure 1. It was also unclear why the authors didn't also apply a statistical cut of (p-value or other). And a different cut-off was applied in figure 2 (here with 2 replicates statistics will be limited).

e. Full supplemental tables of all detected proteins, with replicates, ratios and statistics (compared to controls) should be provided.

2) While the authors nicely optimize the membrane isolation and detergent-based extraction and

also test multiple different antibodies for IP-MS (admittedly this is quite challenging) it was not completely clear how efficient this pull-down is. While the baits are nicely retrieved is there a way of estimating what gets lost during this process (false negatives during the isolation process). This might not be possible, but maybe the authors can comment (this is even suggested on page 12) of the discussion).

3) The authors talk briefly about PTMs of proteins within the SD network, but then never further evaluate. Are PTMs detected? How many of these proteins are glycosylated, considering these are membrane proteins likely most. How will this affect the IP-MS process? It might be useful to search these data with a FragPipe open search to detect PTMs more unbiased(or similar algorithm)?

4) The use of podocyte-specific dox-inducible knockout mice was interesting (animals generated by others and obtained as gifts). Although it was less clear to me what the exact purpose of these experiments was. According to the introduction it was already known that nephrin and podocin mutation can cause proteinuria? Here the authors show that they can 1) delete these proteins (3A-C) and that this leads to proteinuria (here albumin in urine) with different kinetics? It would have been great if the authors used the animals and applied their IP-MS methods to determine remodeling of the protein networks in a time-dependent manner. While this may not be feasible for all three animal models and multiple timepoint following doc-induced target knockdown, such data could provide important data understanding how proteinuria develops and why kinetics is so different. At the moment I feel this figure has limited value. It simply shows/repeats a phenotype that to a certain degree is known.

5) While the drosophila model provides some new insights (some newly identified PPI network partners were tested – drosophila orthologs) it is still somewhat superficial. Why were these three proteins selected? A siRNA drosophila model should easily be scalable to many more targets. If only three are selected, then these should 1) be better justified and 2) more mechanistically interrogated.

6) Figure 5B might be somewhat over-interpreted. Certainly, deletion of the MERTK drosophila ortholog leads to filtration deficits, it's unclear what really goes in in the protein network as a result of this knockdown, since no such data was collected.

7) Figure 4C should be simplified. Just show the individual datapoints and add a line for the median. The box plots add little value.

Overall, this is an interesting paper that without a doubt has some resource value for the renal biology community. Some additional details are required for the proteomics data. The phenotypic/functional data seemed a bit more superficial and especially the dox-inducible mouse models could be utilized in a more meaningful manner. Some additional rationales for candidate selection and mining strategies would likely add transparency for the reader. The filtration model in Figure 5 seems overly simplistic (especially based on the data currently available).

Responses to the reviewers' comments

Reviewer 1

We thank the reviewer for the positive comments on our work and his suggestions for further improvement that were incorporated into the revised manuscript.

In this study the authors investigated the network of the molecules of the kidney podocyte slit diaphragm with high-resolution proteomic analysis. Then, the function of the molecules identified are investigated. The study provides some of important findings for better understanding the structure and function of the slit diaphragm. However, there are some concerns for this paper. The followings may help to improve the manuscript.

Comments:

(1) The advantages of the techniques adopted in this study should be explained more precisely.

We have added respective explanations/details in the Results part (p. 6-8) and the Discussion section (p. 12-14) of the revised ms and also included a new supplemental figure highlighting determination of specifically co-purified proteins by target-normalized ratios (tnR-measure, new Figure S5).

Briefly, our meAP-MS approach (documented in a series of publications, see response to point (2) below) combines affinity-purifications (APs) with a series of (validated (!)) target-specific antibodies (ABs) and quantitative mass spectrometry (MS). MS-derived protein quantification is subsequently used to identify specifically co-purified interactors by means of stringent negative controls (knock-out and target-unrelated IgGs). The resulting *interactomes* are thus hypothesis-free and solely based on protein abundance values – in sharp contrast to most of the previously published work.

(2) Some proteins, which are reported to be components of the slit diaphragm, are not detected with the method in this study. That seems to be an important point for assuring the availability of the method in this study. More explanations are needed.

We have used meAP-MS for comprehensive analysis of 'native membrane proteins' since 2009 (see Schwenk et al., *Science* 2009, *Nature* 2010; Müller et al., *PNAS* 2010; for short review Schulte et al., 2011) and refined it over the last years (Schwenk et al., 2012, 2016 (*Neuron*, *Nature Neuroscience*), Schmidt et al., 2017 (*Neuron*), Schwenk et al., 2019 (*Neuron*), Kollwe et al., 2021 (*Elife*)). The current work uses some methodological improvements (quantification procedure, tnRs) and tools (BELKI) as described in the Methods section. Availability of the meAP-MS-technique should, therefore, not be a concern.

While some of the previously described components of the SD were confirmed in our work (including Neph1, Neph1, Podocin, ROBO2, kinase FYN, MAGI1,2 and Dendrin), others were either found non-specific by the stringent negative controls used in this study (but not used in previous work (!), including ZO-1, IQGA1, SHIP2, spectrins 1, 2) or were not detected by the MS-analysis of our APs (including TRPC6, NCK1, 2), despite the extended linear dynamic range of the method of four orders of

magnitude. It must be noted in this context, that quite a number of commercially offered ABs tested in the course of this work (and used in previous work) were found to lack target-specificity and/or effective binding of the target (see Figure S2).

This is explicitly stated and extensively described in the revised Discussion section (p. 12-14) as requested by the reviewer. In addition, we added a note of caution about false negatives of our approach (p.13).

(3) The findings shown in Figure 4C are not clear enough to discover the function of the molecules. More explanations on the interpretation of the results are needed.

We have extended the functional investigations of newly identified SD-constituents further corroborating their importance for the filtration process; importantly, proteins expressed in the nephrocytes, but not part of the SD (core-interactome), did not affect filtration.

Moreover, in-depth analyses of the SD-network in the conditional KO mice (Figure 4) and the phosphorylation status of its core components uncovered that the (structural) stability of the network is profoundly impacted by phosphorylation of the core constituents further pointing to the significance of kinases (selected in Figure 5B).

These data and findings were added to the revised ms as new Figure 4 and supplementary Figures S7 and S8, together with additional explanations as requested by the reviewer.

Reviewer 2

We appreciate the reviewers' positive comments, as well as his criticism that prompted additional experiments included into the revised manuscript as detailed below in the responses to the reviewer's comments.

In this manuscript, the authors discovered new proteins that interact with key components of the Slit Diaphragm (SD) using proteomic analysis on membrane fractions from glomeruli of rat & mouse kidney. Using antibodies against Nephrin, NEPH1 and Podocin, they identified many (over 30) novel proteins that have not been shown before to be interacting with the SD. Using Drosophila nephrocytes, they tested the homologs of three of these proteins (MERTK, ANPRC & ITM2B) and showed that their fly homologs are needed for the filtration function. The manuscript is well-written, and it is significant to identify many potential new proteins that interacts with SD, and validate three of them in the fly nephrocyte. The findings are potentially significant for the podocyte field. However, there are some flaws in the data interpretation and more discussion / explanation are needed.

Comments:

*1. The authors set the threshold for specific binding as $tnR > 0.2$ or 0.25 in Figures 1 and Figure 2. This threshold is critical for claiming that the newly identified proteins from this study are not false positive. The authors have used *t*-SNE for the separation of true binding vs. non-specific binding. However, the separation doesn't look great in Figure 2C, and it is not clear how the authors determined the $tnR > 0.2$ or 0.25 cutoff.*

The specificity-thresholds (abundance ratios versus controls) used for determination of target-interactors were derived from (i) integrating the information obtained from all antibody APs (each with distinct background characteristics and target purification efficiency), (ii) knowledge on verified interaction partners and (iii) our well documented experience in evaluation of AP-MS experiments (Schwenk et al., 2012, 2016, 2019; Schmidt et al., 2017, Kollwe et al., 2021). The necessary conversion of protein abundance ratios (in APs versus controls) into a target-normalized measure of co-enrichment (tnR , Methods) is now illustrated in a new supplemental Figure S5.

As a rule of thumb any interactor is enriched (together with the target (!)) by a factor of 5-10 and more (over background).

It must be mentioned, that *t*-SNE is primarily used for visualization of the AP results and their consistency (rather than for determination of specificity) and that any protein interacting with the target is placed in close proximity to the target by this method – which is perfectly indicated for the *anti-Podocin* APs in Figure 2. As for most AP-MS experiments, quantitative measures cannot derive a perfectly sharp threshold for determination of specific interaction partners vs proteins belonging to background.

2. Podocin was previously established as a trans-membrane protein associated with the lipid raft and directly binds to Nephrin and CD2AP in lipid rafts (Schwarz et al., JCI 2001). However, in this study, Podocin was described as a cytoplasmic protein, which serves as a “universal cytoplasmic anchor targeting not only the SD, but also integral membrane proteins with diverse functions”. This is contradictory with a well-established understanding of our previous knowledge about Podocin and needs to be explained in more details.

Podocin is a hairpin-like protein attached to the cytoplasmic leaflet of the membrane (with N- and C-termini located intracellularly; eg. Schurek et al., 2014 or Kadurin et al., 2009). The ‘transmembrane domain’ cited in Schwarz et al. is, in fact, an ‘*intramembrane-attachment domain*’; this configuration is in line with the topology of other stomatin family proteins (Goodman et al., 2002) and our own results obtained with both freeze-fracture replicas and epitope-localization by fluorescence microscopy. Different from Schwarz et al. (2001), we find that Podocin is not an exclusive anchor for the SD (interacting with Nephrin and Neph1), but, in addition, acts as a cytoplasmic interactor/linker for a number of SLC-type and other transporters in the plasma membrane of podocytes.

As a consequence, Podocin did not exclusively localize to the SD in freeze-fracture replicas, but rather displayed a broader distribution over the plasma membrane of the podocytes (Fig. 2B). Importantly, this dual role is independently reflected by the newly determined Podocin *interactome* (Fig. 2, Table 2): It shows partial overlap with the Nephrin/Neph1 *interactome* (Fig. 1, Table 1), but also comprises a unique set of proteins including the aforementioned types of transporters that do not interact with Nephrin/Neph1.

This has been discussed in the revised ms (p. 14) and a note on the current knowledge of Podocin has been added to the Results section (p. 7) as suggested by the reviewer.

3. CD2AP was shown to be directly binding to both Podocin and Nephrin. However, it was not found among the proteins identified in this study. Why is that?

CD2AP was not determined as a specific target-interactor in our APs using 5 different (validated) antibodies directed against Nephrin, Neph1 and Podocin. CD2AP may, therefore, either fail to interact with the SD core, or the interaction may be susceptible to the detergent buffers used in this study. A respective note of caution was added to the Discussion section (p. 13).

4. The authors used Drosophila nephrocytes to show that the fly homologs of three proteins (MERTK, ANPRC & ITM2B) identified from this study are required for the filtration function, but didn't explain why these three proteins were chosen. Are these proteins the only ones that are conserved in flies? If not, why choose these three instead of others?

We have extended the functional investigations of newly identified SD-constituents further corroborating their importance for the filtration process. Importantly, proteins expressed in the nephrocytes (and in podocytes), but not part of the SD (core-interactome) did not affect filtration. These additional findings were added to the revised ms as new Figure 4.

The selection of the SD-networks constituents was based on interactors being new (not reported before), showing documented expression (of orthologues) in flies and

displaying interesting links in literature. This rationale has been introduced into the Results section of the revised ms (p. 10).

5. The nephrocyte RNAi experiments lack sufficient demonstration of RNAi knock-down efficiency. Alternatively, the authors could use independent RNAi lines to validate the knock-down results to avoid any off-target effects.

Reliable quantification of mRNA in larval nephrocytes is difficult (if not impossible). We have, therefore, included a series of additional RNAi-mediated knock-down lines into the new Figures 4 as suggested by the reviewer.

6. The model in Figure 5A is mainly based on speculation. The data from this study only support that these proteins are in close range of the key SD proteins (Nephrin, NEPH1 and Podocin). It is not clear to this reviewer how these specific domain binding, inside or outside of the membrane, could be suggested by the results from this study.

The 'structural model' presented in Figure 5A was meant to (i) illustrate the differences in molecular appearance of the SD introduced by the interactome analysis in our work and to (ii) serve as a first working model. In particular, the figure should emphasize the network-character of the newly identified interactors of the SD core-components and its significance for anchoring Nephrin-Neph1 in opposing foot processes. All newly identified network constituents were roughly drawn to scale (using pdb-data and AlphaFold as a basis), details of interaction sites as well as the 'order of interaction(s)' were not intended – as both are currently unknown (as are the primary functions of the majority of the SD network constituents).

This was explicitly clarified in the legend to Figure 5A of the revised ms as requested by the reviewer.

7. The model in Figure 5B is also mainly based on speculation. If the authors believe that MERTK is the key protein that controls SD dynamics, some data to support this idea should be shown. The author mentioned that "Alteration or removal of this signaling will result in defective filtration as observed with the MERTK knock-down and thus eventually lead to proteinuria (Fig. 3).", but this reviewer could not find any MERTK knock-down data in Figure 3. Is this a typo?

The 'functional model' of the SD (Figure 5B) depicting one component, protein kinase MERTK, was meant to (i) illustrate the potential significance of the SD-associated network for the dynamics of the filtration processes and (ii) to provide a first hypothetical model for such dynamics – as a working model. This was explicitly clarified in the figure legend of the revised ms, and the typo rightfully stated by the reviewer, was corrected.

Moreover, prompted by the reviewers' comments we performed an in-depth analysis of the SD-remodeling induced by the Nephrin knockout – which uncovered the importance of phosphorylation for the stability of the SD and the filtration process. More explicitly, the phosphorylated SD (in particular a triple 'phospho-cluster' in the Nephrin C-terminus) appeared markedly more stable than the unphosphorylated form.

These data including quantification of phosphorylated vs unphosphorylated peptides were added to the revised ms as new supplemental Figures S4, S7, S8 and Table S4.

A respective discussion of the findings and their relation to the functional significance of the SD network was added to the Discussion section of the revised ms.

Reviewer 3

We appreciate the reviewers' positive comments, and also his criticism that triggered additional experiments the results of which are included into the revised manuscript as detailed below in the responses to the reviewer's comments.

The manuscript titled "A slit-diaphragm-associated protein network for dynamic control of renal filtration" by Kocylowski and colleagues utilizes rodent kidney tissue extracts and affinity purifications of various known slit-diaphragm-associated proteins. Isolated complexes are then analyzed by LC-MS to generate a protein network of SD-associated protein. Using both genetic mouse models and a drosophila nephrocyte system to evaluate filtration process, functional relevance for several newly identified SD components was demonstrated.

Overall, this is an interesting manuscript that uses rigorous biochemical purification of SD components to establish a slit-diaphragm-associated protein network. This component of the manuscript is certainly novel and without a doubt relevant to the renal community. This part of the paper will have a significant resource value.

A limited number of newly identified proteins are then evaluated in a drosophila functional assay. While this component of the manuscript is significantly briefer, it is appropriate for a more proteomics centric manuscript.

There are a couple of comment the authors should address to improve presentation and clarity of their results.

Comments:

1) Considering this paper is very proteomics centric the authors need to deposit their raw data to a depository. Most importantly the authors need to provide supplemental tables of their filtered mass spec data. This way others can mine their data. I am very surprised this is missing.

a. Along these lines. Figure 1D/Table 1 and Figure 2D/Table 2 need more information to be useful. In the figures the authors should add MaxQuant LFQ values (it seems this algorithm was used). If proteins weren't detected (or imputed) this needs to be shown in this figure as well (for example in the white boxes – I would assume).

b. Why was IP-MS done in triplicates for figure 1 and only in duplicates for figure 2?

c. In the tables the authors should also add the total number of peptides (or spectra) that were detected for each protein reported.

d. It seems a bit unclear (although somewhat described in the methods) why a tnR ration of >0.2 was selected in Figure 1. It was also unclear why the authors didn't also apply a statistical cut of (p-value or other). And a different cut-off was applied in figure 2 (here with 2 replicates statistics will be limited).

e. Full supplemental tables of all detected proteins, with replicates, ratios and statistics (compared to controls) should be provided.

All data were deposited to the PRIDE database (Project Acc.: PXD034356) and supplemental tables (Tables S1-4) are provided that contain all details of our MS analyses (not transmitted to the reviewer in the original version?).

Specific points:

(a) Quantitative evaluation of the MS data was done as detailed in Methods, i.e. MaxQuant was used for extracting peptide intensity information as well as for

calibration of accurate m/z . We further processed these data using in-house developed software for peptide intensity (peak volume (PV)) assignment (Bildl et al., 2012) and evaluated their internal consistency ('expected PVs'; detailed in Methods) to obtain 'weighting factors' for fitting the data (rather than unweighted fitting as in LFQ determination). The obtained molecular abundance values (basically comparable to LFQ values, but more accurate, in particular for quantification from sparse PV data) for the AP-MS datasets used in Figures 1, 2 were added to supplemental Tables S1 and S2. For proteins not detected, abundance thresholds for detection were calculated from the PV data as described in Methods.

Figures 1D, 2D show the consistency and specificity of proteins based on (target-normalized) abundance ratios (in AP versus multiple controls), color-coded as indicated in the figure. We have extended the figure legends to explain this in more detail. Also, white boxes reflecting failure to detect/quantify a protein in the respective AP have been marked with an X (revised Figures 1D, 2D).

(b) The columns in Figures 1D and 2D show the results from distinct AP antibodies used here (3 for Neph1, 2 for Podocin, as indicated in Figs. 1A, 2A), not from triplicates. This is also the reason why statistical measures (such as p-values) were not be provided. In our experience, the use of different antibodies is far more effective to correct for AP-MS errors than triplicate APs that may eliminate technical flaws but actually manifest the shortcomings of individual antibodies.

(c) Supplemental tables providing information on identified proteins, peptides and spectral matches were part of the original Figures S1 and S2; data for Figure S7 (*interactome constituents* after KO induction of the SD core components) were added to the revised ms as new supplemental Table S3.

(d) To derive threshold values for specificity (abundance ratios versus controls) across different antibody APs (each with distinct background characteristics and target purification efficiency) we used (i) tSNE-plots (integrating the information from all AP experiments), (ii) knowledge on verified interaction partners and (iii) our well documented experience in evaluation of AP-MS experiments (Schwenk et al., 2012, 2016, 2019; Schmidt et al., 2017, Kollwe et al., 2021). The necessary conversion of protein abundance ratios (in APs versus controls) into a target-normalized measure of co-enrichment (tnR, see Methods) is now illustrated in a new Supplemental Figure S5.

(e) Quantitative data on all proteins detected in the meAP experiments, as well as in total membranes from KO animals were added the revised Tables S1 (Neph1 interactome), S2 (Podocin interactome) and S3 (membrane fractions from WT and KO mice) as requested by the reviewer.

2) While the authors nicely optimize the membrane isolation and detergent-based extraction and also test multiple different antibodies for IP-MS (admittedly this is quite challenging) it was not completely clear how efficient this pull-down is. While the baits are nicely retrieved is there a way of estimating what gets lost during this process (false negatives during the isolation process). This might not be possible, but maybe the authors can comment (this is even suggested on page 12) of the discussion).

The efficiency of target purification can now be read from the molecular abundance values in supplemental Tables S1 and S2. Since the lowest abundant (verified)

interaction partners were between 0.004-0.04 % of the target, we consider our evaluation to a depth close to the technical limit.

However, our stringent controls and consistency criteria were chosen to minimize false-positives at the cost of potentially missing additional interaction partners (false negatives, that cannot be assessed as rightfully stated by the reviewer).

This has been added as a note of caution to the Discussion section of the revised ms (p. 11).

3) The authors talk briefly about PTMs of proteins within the SD network, but then never further evaluate. Are PTMs detected? How many of these proteins are glycosylated, considering these are membrane proteins likely most. How will this affect the IP-MS process? It might be useful to search these data with a FragPipe open search to detect PTMs more unbiased (or similar algorithm)?

Our MS datasets are routinely searched against databases including phosphorylation and acetylation - without specific enrichment of phospho- or glycopeptides. The respective MASCOT searches revealed a number of unambiguous phospho-sites (see new supplemental Table S4 and Figure S7, after careful manual inspection of the individual MS/MS spectra). For Neph1, a particularly striking phosphorylation cluster was identified, in line with previous work (Rinschen et al., 2015). First quantitative assessment of its phosphorylation status upon Neph1 knockdown (Figure S8) points to a functional dynamic role for maintaining network stability (added to Discussion and revised Figure 5).

Although glycosylation and the hydrophobic nature of proteins may impact ionization of their peptides and detection in MS, the high sensitivity of our MS setup and quantification pipeline did not introduce a noticeable bias against membrane proteins (12-25 % of detected in total and 65-89 % of proteins determined specifically co-purified with their target were integral membrane proteins (according to their annotation in UniProt/Swiss-Prot.), similar to the observations in our previous work.

4) The use of podocyte-specific dox-inducible knockout mice was interesting (animals generated by others and obtained as gifts). Although it was less clear to me what the exact purpose of these experiments was. According to the introduction it was already known that neph1 and podocin mutation can cause proteinuria? Here the authors show that they can 1) delete these proteins (3A-C) and that this leads to proteinuria (here albumin in urine) with different kinetics? It would have been great if the authors used the animals and applied their IP-MS methods to determine remodeling of the protein networks in a time-dependent manner. While this may not be feasible for all three animal models and multiple timepoint following doc-induced target knockdown, such data could provide important data understanding how proteinuria develops and why kinetics is so different. At the moment I feel this figure has limited value. It simply shows/repeats a phenotype that to a certain degree is known.

As correctly stated by the reviewer, Neph1, Neph1 and Podocin were genetically targeted in constitutive knock-out animals (before) which, due to early phenotypes or perinatal lethality, however, could not be used for investigation of SD-composition and/or impact on filtration.

Therefore, we had to generate inducible knock-out animals using a highly efficient tetracycline-inducible Cre-line on the same genetic background for all alleles (and

controls) investigated. This allowed, for the first time, direct comparison between the three core-components on both development of proteinuria and re-modelling of the SD composition.

Data on remodeling of the SD network together with in-depth analysis of the phospho-status were included into the revised ms as text and new Figures S7 and S8 as suggested by the reviewer. Briefly, these data identify the SD constituents that are directly dependent on to the core components (Figure S7, compatible with direct interactions) and they show that the stability of the SD network is profoundly impacted by phosphorylation of Nephrin (Figure S8).

5) While the drosophila model provides some new insights (some newly identified PPI network partners were tested – drosophila orthologs) it is still somewhat superficial. Why were these three proteins selected? A siRNA drosophila model should easily be scalable to many more targets. If only three are selected, then these should 1) be better justified and 2) more mechanistically interrogated.

We have extended the functional investigations of newly identified SD-constituents further corroborating their importance for the filtration process. Importantly, proteins expressed in the nephrocytes (and in podocytes), but not part of the SD (core-interactome) did not impact filtration. These additional findings were added to the revised ms as new Figure 4.

The selection of the SD-networks constituents was based on interactors being new (not reported before), showing documented expression (of orthologues) in flies and displaying interesting links in literature. This rationale has been introduced into the revised ms.

6) Figure 5B might be somewhat over-interpreted. Certainly, deletion of the MERTK drosophila ortholog leads to filtration deficits, it's unclear what really goes in in the protein network as a result of this knockdown, since no such data was collected.

The 'functional model' of the SD (Figure 5B) depicting one component, protein kinase MER, was meant to (i) illustrate the potential significance of the SD-associated network for the dynamics of the filtration processes and (ii) to provide a first hypothetical model for such dynamics – as a working model. This was explicitly clarified in the figure legend of the revised ms.

Moreover, prompted by the reviewers' comments we performed an in-depth analysis of the SD-remodeling induced by the Nephrin knockout – which uncovered the importance of phosphorylation for the stability of the SD and the filtration process. More explicitly, the phosphorylated SD (in particular a triple 'phospho-cluster' in the Nephrin C-terminus) appeared markedly more stable than the unphosphorylated form.

These data including quantification of phosphorylated vs unphosphorylated peptides were added to the revised ms as new supplemental Figures S4, S7, S8 and Table S4. A respective discussion of the findings and their relation to the functional significance of the SD network was added to the Discussion section of the revised ms.

7) Figure 4C should be simplified. Just show the individual datapoints and add a line for the median. The box plots add little value.

For transparency reasons - showing all data points together with mean (filled squares) and error of the mean (bars) – we would prefer to leave the presentation as it is.

Overall, this is an interesting paper that without a doubt has some resource value for the renal biology community. Some additional details are required for the proteomics data. The phenotypic/functional data seemed a bit more superficial and especially the dox-inducible mouse models could be utilized in a more meaningful manner. Some additional rationales for candidate selection and mining strategies would likely add transparency for the reader. The filtration model in Figure 5 seems overly simplistic (especially based on the data currently available).

REVIEWERS' COMMENTS

Reviewer #1 (Remarks to the Author):

The manuscript has been revised well.
The authors properly responded to my comments.
I understand the authors' responses to my comments #2 and #3.

Reviewer #2 (Remarks to the Author):

Most of the concerns have been appropriately addressed.

Reviewer #3 (Remarks to the Author):

This is a strong paper that will have significant resource value to the renal biology community.

The authors have carefully addressed my previous comments. I have no additional concerns. The manuscript should be accepted in Nature Communications.

Response to reviewers

Reviewer #1

The manuscript has been revised well.

The authors properly responded to my comments.

I understand the authors' responses to my comments #2 and #3.

Reviewer #2 (Remarks to the Author):

Most of the concerns have been appropriately addressed.

Reviewer #3 (Remarks to the Author):

This is a strong paper that will have significant resource value to the renal biology community.

The authors have carefully addressed my previous comments. I have no additional concerns. The manuscript should be accepted in Nature Communications.

All comments are very much appreciated